# Global transportation infrastructure exposure to the change of precipitation in a warmer world

Kai Liu [1,2] ✉, Qianzhi Wang [1,3], Ming Wang[1] & Elco E. Koks [4] ✉

Transportation infrastructures are generally designed to have multi-decadal service lives. Transport infrastructure design, however, is largely based on historical conditions. Yet, in the face of global warming, we are likely going to experience more intense and frequent extreme events, which may put infrastructure at severe risk. In this study, we comprehensively analyze the exposure of road and railway infrastructure assets to changes in precipitation return periods globally. Under ~2 degrees of warming in mid-century (RCP 8.5 scenario), 43.6% of the global transportation assets are expected to experience at least a 25% decrease in design return period of extreme rainfall (a 33% increase in exceedance probability), which may increase to 69.9% under ~4 degrees of warming by late-21st century. To accommodate for such increases, we propose to incorporate a safety factor for climate change adaptation during the transportation infrastructure design process to ensure transportation assets will maintain their designed risk level in the future. Our results show that a safety factor of 1.2 would work sufficient for most regions of the world for quick design process calculations following the RCP4.5 path.

Reliable transport infrastructure provides the backbone of international trade and well-functioning economies[1]. Nevertheless, the reliability of transport infrastructure is regularly under threat because of natural hazards. Globally, the multi-hazard risk due to direct damage to road and railway assets is estimated to range between 3.1 billion and 22 billion USD[2]. As a result of the increasing global mean temperature, it is expected that extreme climate events will increase in both intensity and frequency in the future[3–6]. Over the last century, research estimates have shown an average increase in the annual maximum daily rainfall intensity of approximately 6–8% per °C of warming[7]. However, this signal can vary substantially across regions, with some regions experiencing above average increases, whereas other regions may experience decreases in extreme precipitation[8]. And towards the future, this trend is expected to continue[6].

This increased pressure on transport infrastructure may cause a further deterioration of infrastructure assets and increased maintenance and replacement costs[1]. As such, building and maintaining resilient, sustainable, and reliable infrastructure is one of the key targets of Sustainable Development Goal 9. To achieve this goal, low- and middle-income countries need to spend between 0.5% and 3.3% of their gross domestic product annually (from 157 billion to 1 trillion USD) on new transport infrastructure by 2030, plus an additional 1–2% of their gross domestic product to maintain their networks, depending on their ambition and their service delivery efficiency[9]. To successfully achieve Sustainable Development Goal 9 through building climate-resilient and sustainable infrastructure, it is first necessary to better understand the impacts of climate extremes on infrastructure assets. While much work has been done toward understanding the direct

[1]School of National Safety and Emergency Management, Beijing Normal University, Beijing, China. [2]Collaborative Innovation Center on Forecast and Evaluation of Meteorological Disasters (CIC-FEMD), Nanjing University of Information Science & Technology, Nanjing, China. [3]School of Systems Science, Beijing Normal University, Beijing, China. [4]Institute for Environmental Studies (IVM), Vrije Universiteit Amsterdam, 1081 HV Amsterdam, Netherlands. ✉e-mail: liukai@bnu.edu.cn; elco.koks@vu.nl

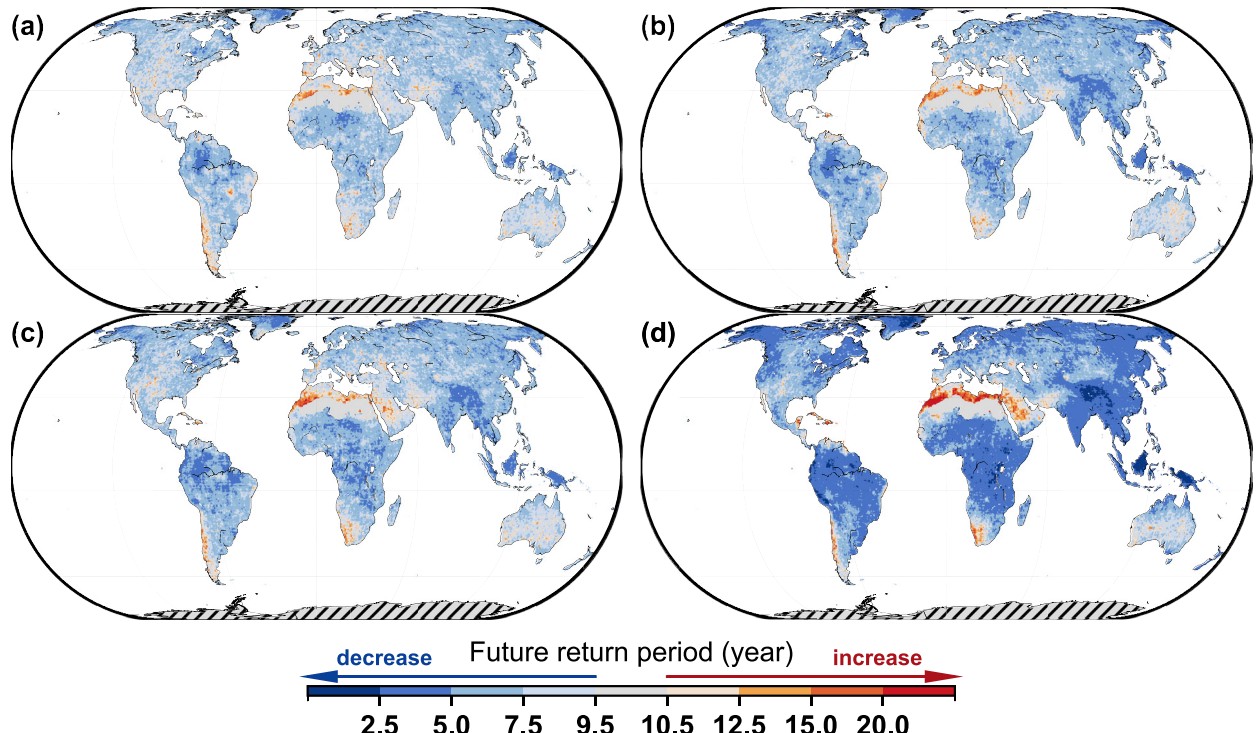

**Fig. 1 | Multi-model median return period for precipitation in the future for 1-in-10-year precipitation compared with the period of 1971–2000. a** mid-21st century (2030–2059) under the RCP4.5 scenario. **b** late-21st century (2070–2099) under the RCP4.5 scenario. **c** mid-21st century (2030–2059) under the RCP8.5 scenario. **d** late-21st century (2070–2099) under the RCP8.5 scenario.

economic damage inflicted on infrastructure assets[2,10], and some work has focused on indirect economic losses resulting from network disruptions[11,12], little work has been done toward understanding how design standards need to change in the future to mitigate the impacts of climate extremes on infrastructure assets[13]. Accordingly, this study examines how the probability of extreme precipitation events may change in the future and where, and by how much, design standards need to change to achieve and maintain reliable and well-functioning infrastructure systems.

Here, we analyze future changes in the global exposure of roads and railways (see Methods and Supplementary Table 1) to precipitation in a warmer world using multi-model projections from the Coupled Model Intercomparison Project Phase 5 (CMIP5). Accordingly, we first estimate the precipitation return period shifts under different global circulation models and then investigate how this change influences the transportation exposure by incorporating the design standards of infrastructure drainage systems. Finally, detailed regional differenced safety factors are suggested for designing transportation drainage systems to cope with precipitation in a changing climate. Our study provides a picture of the magnitude of the threat to transportation infrastructure from climate change at the global scale and helps bridge the knowledge gap between climate science communities, infrastructure communities, and policy makers. This knowledge can also incentivize governments or municipalities to design infrastructure adaptation measures to cope with the risk of anthropogenic warming.

## Results
### Changes in the precipitation return periods
The changes in the precipitation return periods between the present (1971–2000) and two future horizons (2030–2059 and 2070–2099) were analyzed. The magnitudes of the precipitation for certain return periods (for example, 2, 5, 10, 20, 30, and 50 years) in the present day were calculated for each grid. The corresponding return period of the same magnitude precipitation was then computed for the time series

of the future precipitation for each grid under different climate scenarios. The details of the above processes are described in the Methods.

Figure 1 shows an example of how the probability of a 1-in-10-year precipitation event (an event that has a 10% probability of exceedance in any one year) may change in the future. The results for other return periods (Supplementary Figs. 3–7) show similar spatial distributions, even though the magnitude varies. From Fig. 1, we observe that roughly 91.7–94.6% of the global land mass may experience decreasing return periods as a result of global warming. Increasing return periods (decreasing frequency) are primarily observed in northern Africa, southwest of South America, and Saudi Arabia. Under the RCP4.5 scenario, 58.7% (-1.8 degrees of warming in mid-21st century) and 73.8% (-2.5 degrees of warming in late-21st century) of the world is projected to have a return period decrease larger than 25% (i.e., a 1-in-10-year event becomes a less than 1-in-7.5-year event). Under the RCP8.5 scenario, these numbers increase to 71.5% and 86.6%, for mid-21st century (-2 degrees of warming) and late-21st century (-4 degrees of warming), respectively. Greenland, eastern and western North America, northern South America, Central Africa, the eastern Siberian Plateau, Central India, Southwest China, and Southeast Asia are the regions most sensitive to global warming that will face the most significantly shortened return period of precipitation when the global mean temperature increases from the mid-21st century to the late-21st century.

### Exposure analysis
Transportation infrastructures are most often designed to be able to resist a certain return period rainfall event[14]. When the probability of a rainfall event exceeds this design threshold, the excess water may adversely affect the transport assets. This could cause disruption of its usage, or structural degradations such as road surface erosion, reduced bearing capacity, and shortened structural life. More specifically, the infrastructure drainage system is expected to protect

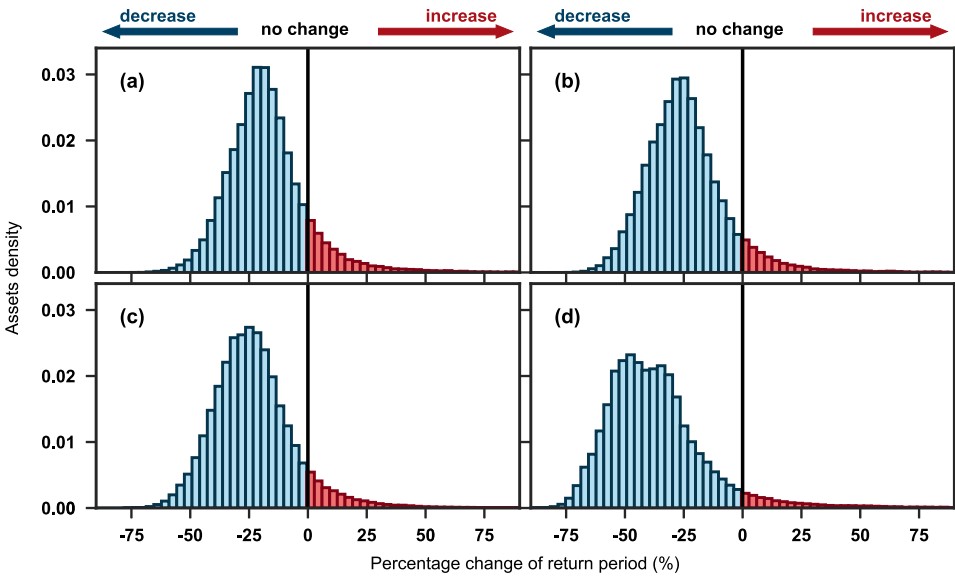

**Fig. 2 | Percentage distribution of global road and railway assets facing changes in the design return periods under different time periods and scenarios. a** mid-21st century (2030–2059) under the RCP4.5 scenario. **b** late-21st century (2070–2099) under the RCP4.5 scenario; (**c**) mid-21st century (2030–2059) under the RCP8.5 scenario. **d** late-21st century (2070–2099) under the RCP8.5 scenario. Red and blue color denotes the percentage of transportation assets facing an increase in precipitation return periods, respectively.

transportation infrastructure against rainfall events with return periods smaller than the design return period. If, under global warming scenarios, the return period based on the historical precipitation record becomes shorter (i.e., the frequency increases), infrastructure systems may become less reliable than anticipated. Figure 2 shows the percentage distribution of road and railway assets facing changes in the design return period under different time periods and scenarios. Different drainage system design standards (i.e., design return periods) are assigned for 218 countries in different income groups as well as different assets (see Methods and Supplementary Table 1). We find that the design return periods of nearly 88.4–94.6% of global transportation assets will become shorter relative to the historical period of 1971–2000, with average decreases of 24.6% and 34.2% (for the mid-21st century and late-21st century, respectively, given the mean of RCP4.5 and RCP8.5) and standard deviations of 13.3% and 14.9% (for the mid-21st century and late-21st century, respectively, given the mean of RCP4.5 and RCP8.5). The percentage distribution in the late-21st century shifts to the left compared with that of the mid-21st century; in particular, the 0–25% interval moves significantly toward the 25–50% interval and the proportion above a 50% change increases as well (that is, the exceedance probability of such an extreme event is more than doubled).

The exposure of infrastructure assets to changes in precipitation is affected by two factors: the spatial distribution of the assets and the change in the design return period at a given location. Figure 3 shows the spatial distribution of the absolute exposure of global transportation infrastructure, which is expressed by the sum of all road and railway assets facing a more than 25% decrease in the precipitation design return period within a grid of approximately 25 km × 25 km (see Methods). The individual exposures for each road and railway asset category can be found in Supplementary Fig. 8–13. The distributions of the absolute exposure are similar under the RCP4.5 and RCP8.5 scenarios. Under the RCP4.5 scenario, 6.8 million km (mid-21st century) and 11.0 million km (late-21st century) of global road and railway assets will be exposed to more frequent extreme precipitation (i.e., a reduction ratio in the design return period of more than 25%), representing 28.8% and 46.6% of the global land transport infrastructure, respectively. Under the RCP8.5 scenario, the exposed transportation assets increase to 10.3 million km (mid-21st century) and 16.5 million

km (late-21st century), accounting for 43.6% (mid-21st century) and 69.9% (late-21st century) of the global transportation assets. Regions that experience high absolute exposure are clustered in eastern North America, northern Western Europe, Central Europe, and East Asia; this can primarily be attributed to the dense transportation networks in these regions. Although the infrastructure density dominates the absolute exposure, when we further investigate the rankings of countries (see Supplementary Data 1), we find that both the infrastructure density and the precipitation change contribute. Sweden and Greece, for example, are ranked respectively 20th and 39th place for the total length of road and railway assets. Under the mid-century RCP4.5 scenario, however, their rankings are 9th (Sweden) and 84th (Greece) in absolute exposure, unveiling the contribution of precipitation change.

Figure 4 shows the relative global transport infrastructure exposure; that is, the ratio of the absolute exposure to the total assets (see Methods). Compared to absolute exposure, the relative exposure can highlight countries with fewer assets but are strongly affected by precipitation change. These countries are usually less developed countries and may be overseen when exploring the results for absolute exposure due to their relatively low amount of transportation assets. Under the RCP4.5 scenario, grids with an exposure ratio larger than 80% (see Methods) account for 22.5% (mid-21st century) and 40.5% (late-21st century) of the total exposed grids. We find that the relative exposure varies widely across the globe. Under the RCP8.5 scenario, the percentage of grids with an exposure ratio larger than 80% increases to 36.6% (mid-21st century) and 69.8% (late-21st century). These highly exposed areas are concentrated in the eastern and western United States, southeastern South America, Central and Northern Europe, Southwest China, and Southeast Asia for the mid-21st century, and extend to Central Africa, Central India, and Southeast Australia for the late-21st century. Interestingly, we do find some countries experience a low absolute exposure but a high relative exposure. In Panama, for example, 99% of the grids will face an exposed ratio of over 80% in the late-21st century under the RCP8.5 scenario, making a large proportion of the transportation infrastructure in Panama highly vulnerable to rising temperatures. Yet, Panama only has a total length of assets of 6539 km and ranks 107th in absolute exposure in the late-21st century under the RCP8.5 scenario (see Supplementary Data 1).

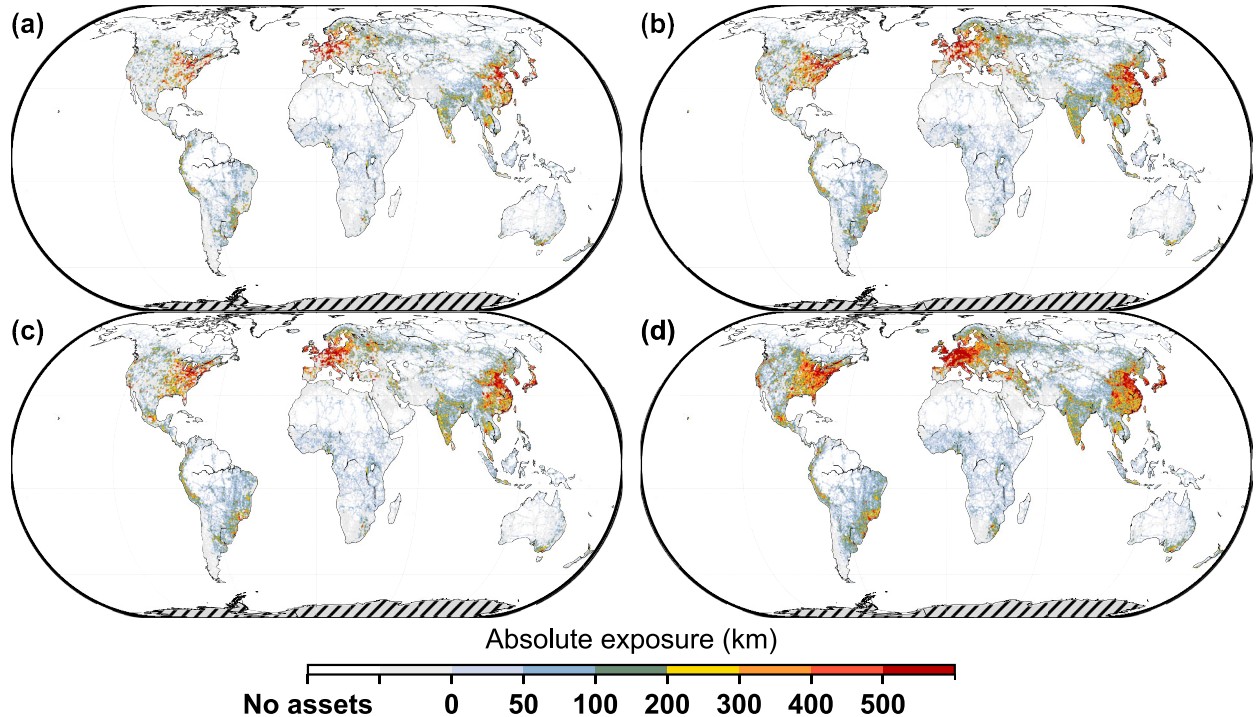

**Fig. 3 | Spatial distribution of absolute exposure of global road and railway assets under different time periods and scenarios. a** mid-21st century (2030–2059) under the RCP4.5 scenario. **b** late-21st century (2070–2099) under the RCP4.5 scenario. **c** mid-21st century (2030–2059) under the RCP8.5 scenario. **d** late-21st century (2070–2099) under the RCP8.5 scenario. Results are shown in a grid size of approximately 25 km × 25 km. The absolute exposure is defined as the total length of road and railway assets within a grid exposed to a more than 25% decrease in the design return period in future.

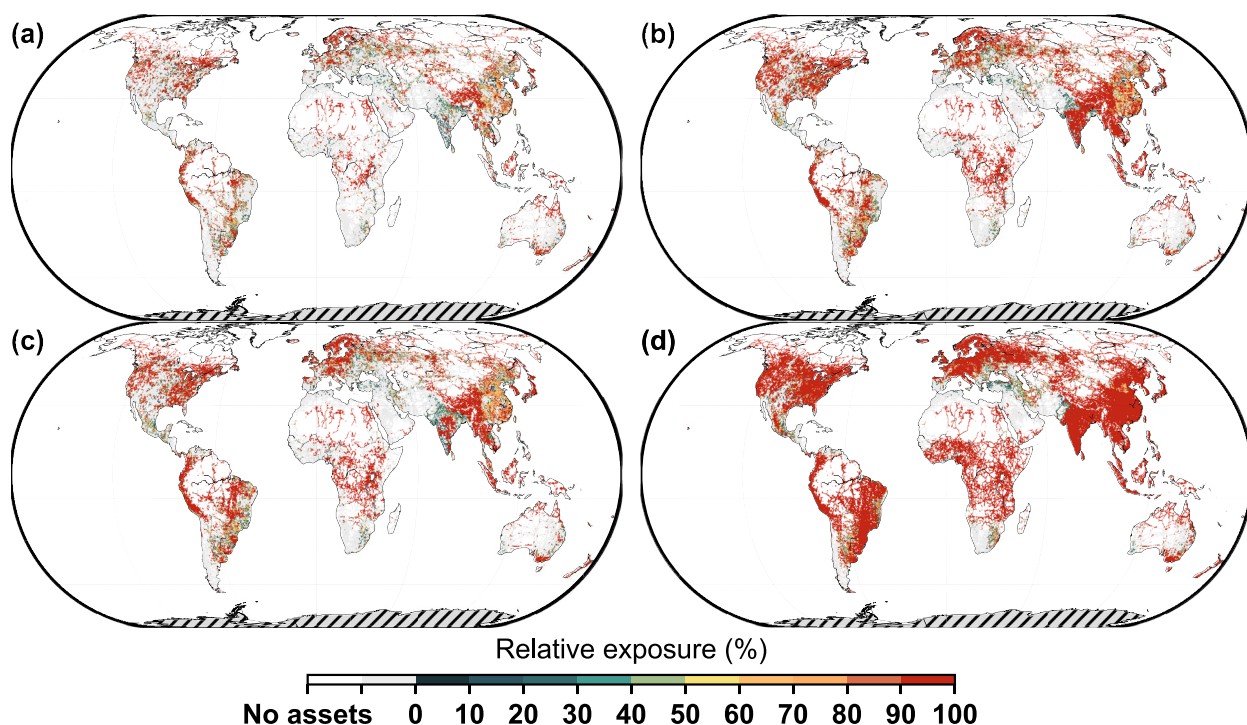

**Fig. 4 | Spatial distribution of relative exposure of global road and railway assets under different time periods and scenarios. a** mid-21st century (2030–2059) under the RCP4.5 scenario; (**b**) late-21st century (2070–2099) under the RCP4.5 scenario; (**c**) mid-21st century (2030–2059) under the RCP8.5 scenario; and (**d**) late-21st century (2070–2099) under the RCP8.5 scenario. Results are shown in a grid size of approximately 25 km × 25 km. The relative exposure is defined as the ratio of the absolute exposure to the total assets within a gird.

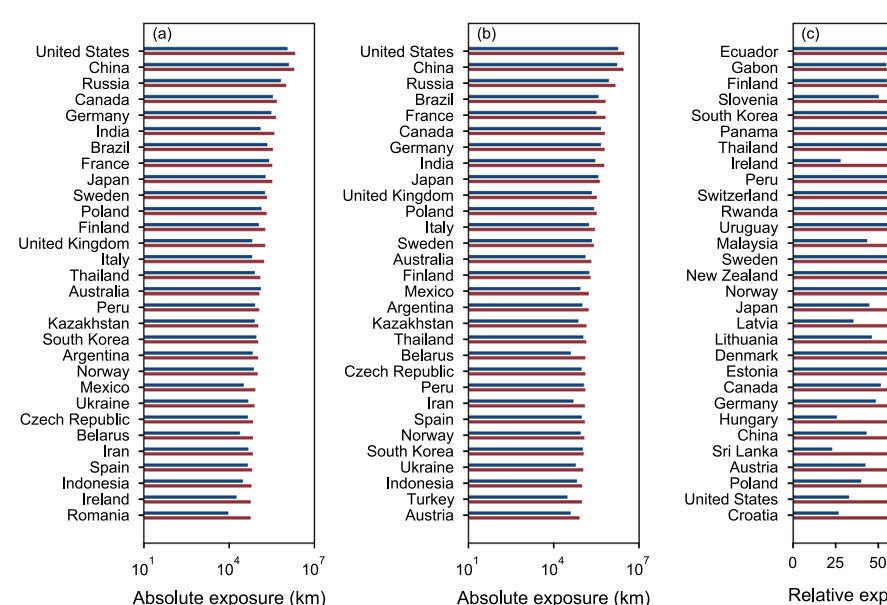
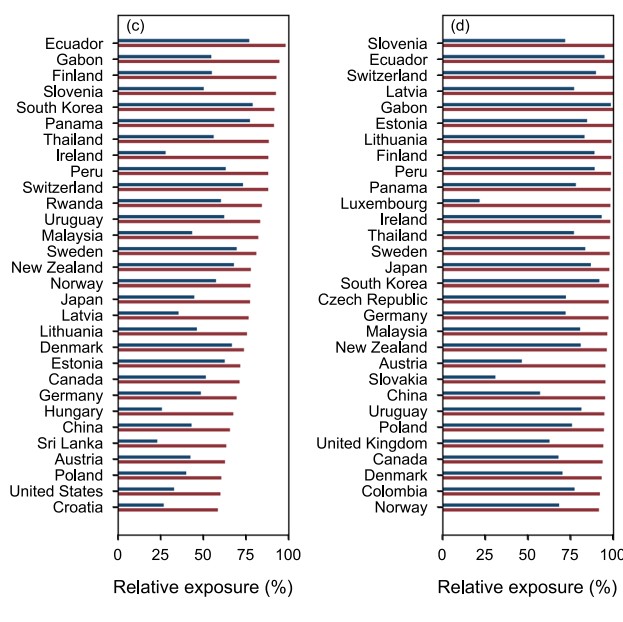

**Fig. 5 | Ranking of countries in absolute and relative exposure of road and railway assets under different time periods and scenarios. a** ranking of countries in absolute exposure under the RCP4.5 scenario. **b** ranking of countries in absolute exposure under the RCP8.5 scenario. **c** ranking of countries with a total asset length of more than 5000 km in relative exposure under the RCP4.5 scenario. **d** ranking of countries with a total asset length of more than 5000 km in relative exposure under the RCP8.5 scenario. The results are presented based on multi-model median projection.

Figure 5 shows the top-30 countries with the highest absolute and relative exposures under the RCP4.5 and RCP8.5 scenarios. The top-30 countries that experience the highest absolute exposure for the RCP8.5 and RCP4.5 scenarios are very similar, with an overlap of 28 countries. For the relative exposure, there is an overlap of 25 countries. Eleven countries exhibit high risk in both absolute and relative exposure under both the RCP4.5 and RCP8.5 scenarios, such as China, Canada, Germany, Japan, Sweden, and Poland.

The United States ranks first in absolute exposure due to its high density of transport infrastructure assets and increased frequency of extreme rainfall events, especially along the East and West coasts. Under the RCP4.5 scenario, the number of exposed assets in the United States will reach 1.14 million km (mid-21st century) and 2.09 million km (late-21st century), accounting for 32.7% (mid-21st century) and 60.1% (late-21st century) of its total assets. China ranks second in absolute exposure, with 1.27 million km (mid-21st century) and 1.94 million km (late-21st century) of exposed roads, accounting for 42.9% (mid-21st century) and 65.5% (late-21st century) of its road and railway assets. A total of 15 European countries and 8 Asian countries can be found in the top-30, including China, Russia, Germany, France, India, and Japan, etc. From the mid-21st century to the late-21st century, the largest impacts appear in Romania, Ireland, and India, resulting in increases of 509%, 216%, and 206%, respectively, in the absolute exposure. Regarding the relative exposure, under RCP4.5 scenarios, a total of 15 European countries made the list, with Finland showing the highest relative exposure, ranking third globally at up to 54.9% (mid-21st century) and 92.7% (late-21st century) of assets exposed. Six Asian countries, South Korea, Thailand, Malaysia, Japan, China, and Sri Lanka, made the list.

**Safety factor for climate change adaptation**
Most transportation infrastructures are designed and built based on the assumption that precipitation will resemble historical patterns. As shown in our study, change of precipitation in a warmer world would make infrastructures assets in most regions of the world more exposed than they used to be, resulting in a potential decrease in the network resilience and reliability because of the increased probability of network disruptions resulting from damaged or temporarily dysfunctioning assets. We therefore propose a safety factor for climate change adaptation (see Methods) for upgrading existing infrastructure or when designing new infrastructure. This safety factor should ensure that the designed level of acceptable risk to be maintained under future warming scenarios. Figure 6 shows the spatial distribution of the safety factor, which represents the ratio of the future precipitation intensity corresponding to the design return periods of various transportation assets to the precipitation intensity in the baseline period. For example, the safety factor for climate change adaptation is 1.5 if the precipitation intensities of the design return period are 20 mm and 30 mm for the current state and the future scenario, respectively. The safety factor for each road and railway asset category can be found in Supplementary Figs. 16–21. Figure 6 shows that under the RCP4.5 scenario, we attribute a safety factor of 1–1.5 for 90.6% (mid-21st century) and 92.1% (late-21st century) of the global land mass, while 0.10% (mid-21st century) and 0.15% (late-21st century) of the global land mass have a safety factor larger than 1.5. These areas primarily being concentrated in Tibet, Central India, and the Andes (South America). Under the RCP8.5 scenario, 91.2% (mid-21st century) and 90.7% (late-21st century) of the global land mass have a safety factor of 1–1.5, while 0.19% (mid-21st century) and 1.7% (late-21st century) of the global land mass have a safety factor of over 1.5. Under RCP8.5, these areas are primarily being concentrated in India, Southwest China and the Indo-China Peninsula, East Africa, the Andes in South America, and north of 50°N.

Additionally, we found that areas with a high safety factor do not completely coincide with areas with a high degree of return period shortening (Section 2.1). This implies that a small increase in precipitation intensity could make a large difference in an area where design standards for infrastructure are already high and where we observe a relatively large change in the design return periods. In addition, we performed a sensitivity analysis to assess changes in the safety factor while considering uncertainties in the design standards (see Methods). Assuming higher design standards (Supplementary Fig. 22), under the RCP4.5 scenario, 87.7% and 90.9% (mid-21st century and late-21st century, respectively) of the global land mass have a

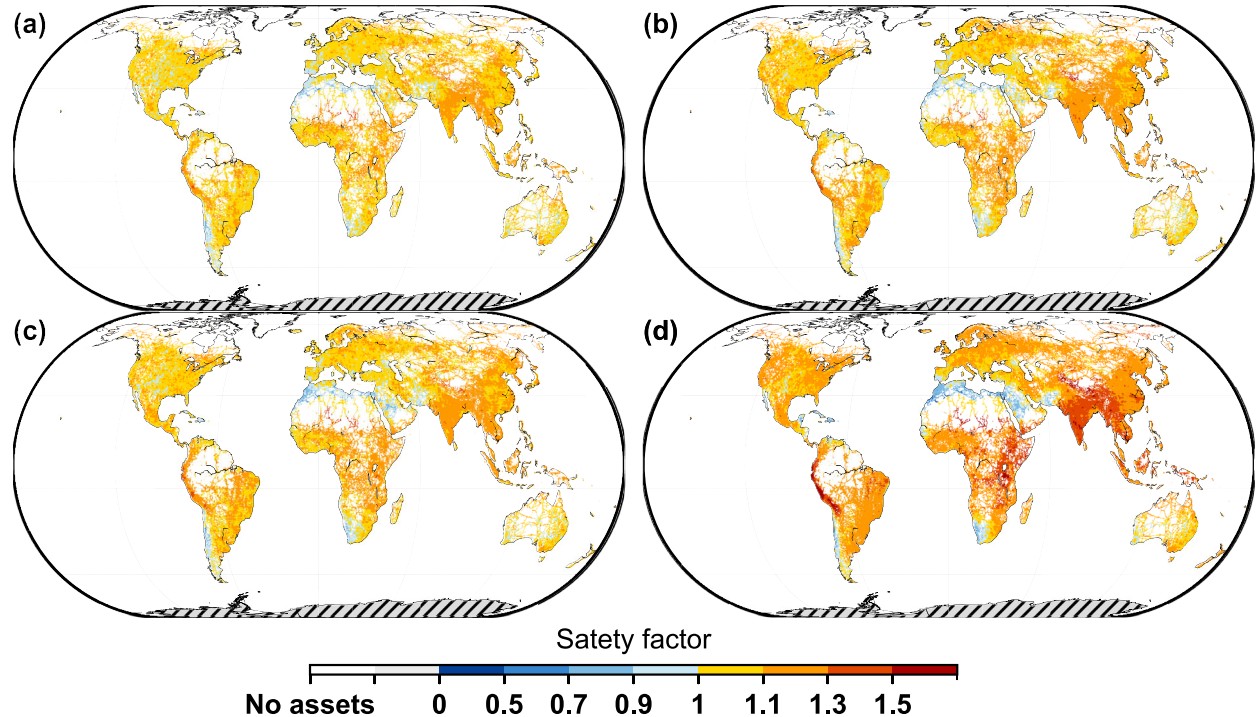

**Fig. 6 | Safety factor for climate change adaptation with respect to current design under different time periods and scenarios. a** mid-21st century (2030–2059) under the RCP4.5 scenario. **b** late-21st century (2070–2099) under the RCP4.5 scenario. **c** mid-21st century (2030–2059) under the RCP8.5 scenario. **d** late-21st century (2070–2099) under the RCP8.5 scenario. Results are shown in a grid size of approximately 25 km × 25 km. A value of 1 indicates no changes in the designed precipitation, a value of 1.2, for example, means a factor of 1.2 is suggested to apply to the designed precipitation intensity.

safety factor between 1 and 1.5, while 0.05% and 0.10% (mid-21st century and late-21st century, respectively) of the global land mass have a safety factor larger than 1.5. This indicates that the safety factor is quite robust.

When considering a safety factor for climate change adaptation, we assume that transportation assets are expected to maintain their designed failure standards towards the future. Under the RCP4.5 scenario, a safety factor of 1.2, while approximate, will work sufficient for most regions of the world for quick design process calculations (87.6% and 82.0% for the mid-21st century and late-21st century, respectively). Under the RCP8.5 scenario, however, these two values are 83.6% and 45.5%, respectively. This highlights the importance of following the RCP4.5 path, which aims to stabilize radiative forcing at 4.5 W m$^{-2}$ in the year 2100.

## Discussion
In this study, we analyzed the exposure of global transportation assets to changes in the precipitation design return period under different future warming scenarios. The change in the extreme precipitation return period challenges the adaptability of the designed return periods of infrastructure drainage facilities. We show that roughly 91.7–94.6% of the global land mass is projected to experience a decrease in their return periods (viz. an increase in exceedance probability). Greenland, eastern and western North America, northern South America, Central Africa, the eastern Siberian Plateau, Central India, Southwest China, and Southeast Asia are regions that will experience some of the most pronounced precipitation return period changes. Nearly 88.4% (mid-21st century, RCP4.5 scenario)–94.6% (late-21st century, RCP8.5 scenario) of global transportation assets will be exposed to precipitation with increased frequencies. Under the RCP4.5 scenario, a total of 6.8 million km (mid-21st century) and 11.0 million km (late-21st century) of global transportation assets will be exposed to more frequent extreme precipitation (decrease in design return period larger than 25%). In relative

terms, 22.5% (mid-21st century) and 40.5% (late-21st century) of grids have an exposure ratio of over 80%. Under the RCP8.5 scenario, absolute exposure reaches up to 10.3 million km (mid-21st century) and 16.5 million km (late-21st century), with the proportion of highly exposed grids increasing to 36.6% (mid-21st century) and 69.8% (late-21st century) compared with the RCP4.5 scenario. The results demonstrate that realizing the RCP4.5 scenario would robustly reduce the exposure of transportation to precipitation extremes, compared with RCP8.5. Regions that will experience the highest absolute exposure are distributed in eastern North America, northern Western Europe, Central Europe, and East Asia. This can primarily be explained by the high density of the transportation infrastructure in these regions. Of these regions, the United States will experience the largest exposure because of its high asset distribution in areas with shortened return periods (on both the East and West coasts).

A safety factor for climate change adaptation, applied to the designed precipitation intensity (i.e., the precipitation intensity corresponding to the designed return period) determined by historical stationary records, provides a practical and useful tool for approximating climate change effects. A change of 1.2 gives a good approximation of the climate change effect for most regions of the world under the RCP4.5 scenario, and provides asset manager a clear guidance for their designs with future planning horizons. In particular, India, Southwest China and the Indo-China Peninsula, East Africa, and the Andes in South America require higher safety factors.

While this study did not assess the feasibility of adaptation measures to reduce the impacts of extreme events, several measures are possible to reduce damage to roads and railways, of which updating drainage systems is the most prominent. Updating or implementing a road or railway drainage system, however, will substantially increase the cost of the required infrastructure for a specific location; this is particularly the case in low income countries. According to a case study of a 1.7-km$^2$ street in China, it is estimated that approximately

0.7 million USD is needed to improve its drainage capacity from that for a 1-in-2-year event to that for a 1-in-5-year event[15]. To assess whether it is cost-effective to implement such measures, the risk of a transportation asset during its design life with respect to future climate change needs to be analyzed and a cost-benefit analysis is required. Note that the benefits will be much larger if we consider not only the direct damages but also indirect losses caused by transportation disruptions. In addition to increasing design standards, natural solutions and green infrastructure can be relatively cost-effective strategies[16–18] that can be integrated with existing drainage systems to reduce the impacts of excessive precipitation. Optimal risk reduction strategies could consist of a mixture of different measures.

To conclude, we emphasize that it is essential to acknowledge uncertainties when designing resilient infrastructure for the future. In particular in the case of infrastructure, which generally has a long lifespan, it is essential to incorporate potential future changes in the exposure to maintain the reliability of the infrastructure over its multidecadal service lifespan given growing climatic uncertainty. Our findings underscore the differences in the impacts of climate change on transportation assets in different countries and provide a global view of hotspots with high levels of exposure to extreme precipitation. Such information is essential in facilitating decision-making regarding the prioritization of regions for adaptation to more extreme rainfall because it is too expensive to increase drainage design standards everywhere. Identifying the regions most exposed will allow for more efficient and optimal spending of the available funds. When moving toward the implementation of individual measures, detailed analyses need to be performed using local models and data.

## Methods
Future extreme precipitation changes were assessed on the basis of two aspects: frequency and intensity. To quantify the transportation assets exposed to extreme precipitation, we used the return periods to describe the change between the mid-21st century and late-21st century warming and the baseline period. Because detailed design information for transportation drainage systems is not available for each country, we assumed that these systems were designed or updated based on the precipitation record for the period 1971–2000 used in this study, i.e., the baseline period.

A return period indicates the recurrence interval; this is an average time period between the occurrences of two events. To determine the precipitation intensity of different return periods, we fit distribution functions of the annual maximum daily precipitation (RX1D) for the baseline and future time periods[19,20]. RX1D is defined as the annual maxima of the daily precipitation amount per year [mm/day][8,21,22]. We fit the RX1D-return period distribution for both the historical condition and the future warming scenarios. Using these distributions, we estimated the precipitation intensity corresponding to the return period of the contemporary asset drainage design in the future distribution function to calculate the return period under the same precipitation intensity. Based on this, we developed an exposure analysis to calculate the number of assets facing significant return period changes. To describe how the precipitation intensity under the contemporary design return period would change in the future, we introduced a safety factor for climate change adaptation to express the ratio of the precipitation intensity in the return period of the contemporary design to that in the same return period in the future. Supplementary Fig. 1 illustrates our methodology.

### Climate data
We used the NASA Earth Exchange Global Daily Downscaled Projections (NEX-GDDP) dataset. This dataset is comprised of downscaled climate scenarios for the globe derived from general circulation model runs conducted under CMIP5[23]. The spatial resolution of the dataset is 0.25° (approximately 25 km × 25 km). See Supplementary Table 2 for the models we used. Compared with the coarse resolution of CMIP5, the NEX-GDDP data show a better ability to simulate extreme precipitation.

We used the multi-model median projected changes to express the spatial distribution; this is a relatively robust method to prevent the overall results from being affected by a single model estimation being too high or too low and to ensure that at least half of the models support changes in the same notation[20].

### Infrastructure data
OpenStreetMap (OSM) is an open-access and free geographic database of global geographic roads, which is collected, edited, and improved by a large number of volunteers. In particular, for roads and railways, OSM is now at a high level of completion and is considered to be the most consistent and complete global database of its type and has been widely used in various studies[2,24,25]. Based on the OSM classification of roads, we chose the five most important categories of roads, i.e., (1) motorways, (2) trunks, (3) primary roads, (4) secondary roads, and (5) tertiary roads. Following the OSM definitions, these categories are interpreted in the following manner: (1) A motorway is a restricted access major divided highway, normally with two or more running lanes plus an emergency hard shoulder. (2) Trunks are the most important roads in a country's system that are not motorways. Such roads need not be divided highways. (3) Primary roads are the next most important roads in a country's system and often link larger towns. (4) Secondary roads are the next most important roads and often link towns. (5) Tertiary roads are the next most important roads and often link smaller towns and villages. Some small roads are not included, such as footways, because it is often uncertain whether a drainage design was considered when such footways were constructed. Additionally, the largest gaps in the OSM data are primarily in the lower tier roads, including footways and residential roads. For railways, we selected rails with full-sized passenger or freight trains in the standard gauge for the given country or state.

The road and rail data used in this study were downloaded from the free Geofabrik download server, which is usually updated daily. Our data were updated as of May 6, 2021. The data consist of linear vector data containing spatial positions. Supplementary Table 3 gives a detailed description and the length of the different categories of transportation assets, and their spatial distribution is given in Supplementary Fig. 2.

### Base design return period for different infrastructure assets
Because countries with different development levels have different design drainage capabilities for their transportation infrastructure, we divided the countries into four categories (e.g., high income, upper middle income, lower middle income, and low income), based on the four development standard levels classified by the World Bank. According to Kuznet's theory[26] and the work of Koks et al.[2], low-income countries usually cannot invest much in infrastructure construction, nor can they guarantee expenditures for disaster recovery. Lower middle-income countries can invest relatively more in infrastructure construction but cannot invest much in their disaster recovery capacity. Upper middle-income and high-income countries can invest significant amounts in both their infrastructure construction and their disaster recovery capacity, with higher investments possible in high-income countries. Accordingly, we assigned different design return periods for the drainage systems in countries with different income groups (see Supplementary Table 1). However, we admit that large uncertainty exists in the design standards across the world. Additionally, even though design standards are generally relatively consistent throughout a single country, for some vast countries, such as the United States, the consistency in the design life can be determined at the state or federal level. Furthermore, how well infrastructure is maintained and its deterioration rate differ across the world and across a given country. These factors were not considered in this study given

the global scale of the analysis and the lack of data availability. Therefore, this study should be considered as a starting point for the identification of global hotspots, from which more detailed analyses can be performed using local data.

While there are differences between countries with respect to their design standards, the same can be said concerning the different types of assets: different infrastructure types have different design drainage capabilities. In this study, we divided the assets into three types: (1) railway; (2) motorway/trunk/primary/secondary; and (3) tertiary. Railways tend to have the most strictly designed infrastructure, followed by motorway/trunk/primary/secondary, which are usually built using similar engineering design standards[27–30]. The third type, tertiary roads, tends to have uneven quality and may not include professional drainage[31,32]. The assumptions concerning the design return periods of the drainage systems of different types of transportation infrastructure are provided in Supplementary Table 1. We assumed two different design return period levels: higher design return periods and lower design return periods.

### Fitting function for calculating precipitation under different return periods

To obtain the daily precipitation of each grid for the different return periods, the generalized extreme value distribution (GEV) was used. The GEV distribution is parameterized with three parameters: location ($\mu$: describing the center of the distribution); scale ($\sigma$: describing the deviation around the mean); and shape ($\xi$: describing the tail behavior of the distribution)[33]. We can fit a sample of extremes to the GEV distribution to obtain the parameters that best explain the probability distribution of the extremes. In this study, the GEV parameters were estimated by fitting the annual maximum daily precipitation series for each grid. A Kolmogorov–Smirnov test was used to test the fitting degree of the GEV distribution at a significance level of 5%[34]. The L-moments method was used to estimate the GEV distribution parameters (based on the Python package "lmoments3").

### Estimation of exposure

We calculated the exposure of the assets at the grid scale. A grid size of approximately 25 km × 25 km was chosen according to the resolution of the climate models. The transportation infrastructure consists of vector data, and its length in each grid was counted. We assumed that, when there was a more than 25% decrease in the design return period, the transportation assets could be affected by the inadequate drainage condition. The absolute exposure ($AE$) is defined as the total number of transportation assets exposed to a more than 25% decrease in the design return period in the future. We calculated the change in the return period for each type of asset separately and then added the exposed length of the different types of assets to obtain the absolute exposure. The relative exposure ($RE$) is defined as the ratio of the absolute exposure to the total assets within a gird. $AE$ and $RE$ can be calculated using Eqs. (1) and (2), respectively.

$$AE_g = \sum_{i}^{n} EL_{i,g} \tag{1}$$

$$RE_g = \frac{AE_g}{TL_g} = \frac{1}{TL_g} \times \sum_{i}^{n} EL_{i,g} \tag{2}$$

Here, $AE_g$ is the absolute exposure of the grid $g$; $RE_g$ is the relative exposure of the grid $g$; $i$ is the type of asset, with a total of $n$ types; $EL_g$ is the length of the assets exposed to a 25% decrease in the design return period in the future within the grid $g$; and $TL_g$ is the total length of the assets within the grid $g$.

According to Eq. (2), if we assume that there are 5 km of primary roads and 15 km of tertiary roads in a grid, if the primary road experiences more frequent extreme precipitation, i.e., a 25% decrease in the design return period in the future, while the secondary roads do not, then the absolute exposure is 5 km and the relative exposure is 5 km/(5 + 15) km = 25%.

### Estimation of the safety factor for climate change adaptation

We used a safety factor for climate change adaptation to describe the ratio of the future extreme precipitation to the contemporary extreme precipitation, as expressed in Eq. (3).

$$SF_{g,i} = \frac{pre_{g,i}^{f}}{pre_{g,i}^{p}} \tag{3}$$

$$SF_g = \frac{1}{n} \times \sum_{i}^{n} SF_{g,i} \tag{4}$$

Here, $SF_{g,i}$ is the safety factor of the $i$th type of asset in the grid $g$; $pre_{g,i}^{f}$ is the future precipitation corresponding to the design return period of the $i$th type of asset in the grid $g$; $pre_{g,i}^{p}$ is the contemporary precipitation corresponding to the design return period of the $i$th type of asset in the grid $g$; and $n$ is the number of asset types.

## Data availability

NEX-GDDP dataset is available at https://www.nccs.nasa.gov/services/data-collections/land-based-products/nex-gddp. The four development standard levels classified by the World Bank are available at https://datahelpdesk.worldbank.org/knowledgebase/articles/906519. Global transport infrastructure data are available at https://download.geofabrik.de/. The source data generated in this study[35] have been deposited in the https://doi.org/10.5281/zenodo.7711820.

## Code availability

The Python package "lmoments3" is available at https://github.com/OpenHydrology/lmoments3. The codes generated this study[36] are available at https://github.com/Vapson/InfExposure.

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

## Acknowledgements

This work was supported by the National Natural Science Foundation of China [grant number 41771538]. The financial support is highly appreciated. Climate scenarios used were from the NEX-GDDP dataset, prepared by the Climate Analytics Group and NASA Ames Research Center using the NASA Earth Exchange, and distributed by the NASA Center for Climate Simulation (NCCS). The data support is highly appreciated.

## Author contributions

K.L. developed the original idea. K.L., M.W., and E. E. K. contributed to the study design. K.L., Q.W., and E.E.K. conducted the analysis. K.L. wrote the original manuscript. K.L, Q.W, M.W., and E.E.K. contributed to scientific interpretations of the results.

## Competing interests

The authors declare no competing interests.
