## [Peer Review File · Nature Communications]

REVIEWER COMMENTS

Reviewer #1 (Remarks to the Author):

A global perspective on transport infrastructure design life and how these need to be rethought in the face of climate change – particularly, increasing precipitation projections – was enjoyable to read and think about. The paper is generally clearly written, with the authors providing simple and effective explanations of terms and concepts. However, I recommend some further editorial attention to sentence structure and grammar as some sentences were difficult to understand.

I can only comment on the transport infrastructure aspects of the paper and not the climate science as the latter is out of my field of expertise. To this end I do have some questions and comments for the authors.

1. I would encourage authors to provide more details about how the transportation mapping work was done for this research, in order to facilitate reproduction. For example, were all the roads (as per the classification from openstreetmaps) in urban regions represented? Also, I believe the analysis was done by assessing roadways within grid cells but more detail could be provided – why was 25x25km chosen, and how is the use of one size appropriate for both highly dense urbanized areas versus remote regions? What GIS software or coding platform was used to do the analysis? Related to this is the computational cost of running the analysis for the entire world at this gridding size.

2. The analysis and results entirely depend upon the (current) design periods assumed for each of the three country categories (low, upper middle and high income) and their 3 categories of infrastructure. Although highly coarse, given the global scope of the analysis I can appreciate why this was adopted. However, infrastructure design standards are often particular to jurisdictions (i.e., metropolitan regions, counties, states) in many countries, each with their particular populations, infrastructure characteristics and climates. Thus, is design life relatively consistent throughout a country? Between countries in a region of the world? Given the heavy reliance of the results on assumed design life, I believe some background and discussion on design life and variations throughout the world would be very helpful to have better perspective on the results, and to understand their limitations with respect to applicability and interpretation. In addition, who is the intended audience for this paper (asides academic researchers) – would it be an international organization like the World Bank? Or individual nations? Or, say, individual municipalities that must make changes to design life? I don't believe this analysis has relevance to the latter, and thus I believe it would be nice to have some discussion of this point. On a related note, on page 5, referring to design return period "historical period 1971–2000," what exactly does this mean? Would this be the assumed return period for each income group country for each type of asset?

3. The authors do point out that the results of absolute exposure metric are influenced by both network density and change in precipitation design return period. It seems this metric could be much more useful if the authors could provide some benchmark (map?) of network density, because as it stands it looks to be far more a reflection of network density rather than change in return period.

4. I believe the assertion in the first line of 2nd paragraph, p 15 should be revised. The paper is illuminating about where in the world assets may be at greatest risk, but standards on how to prioritize drainage system upgrades seems a specifically jurisdictional level issue that I don't believe this work is able to address (they are at differing scales). This should be a relatively simple editorial fix.

Overall I enjoyed thinking about the ideas and perspective this paper brought forward.

Reviewer #2 (Remarks to the Author):

The study's strength is that it takes a global look at infrastructure using open street map to extract roads and railways and a downscaled 25 km climate product that is translated using the GEV method to design precipitation and combines. So nice datasets, neatly set up. The manuscript is well written with just a need for a minor grammar edit and the figures are appropriate with a need for slightly more descriptive captions.

The main problems stem from a lack of understanding regarding hydrologic and hydraulic engineering as well as that role in transportation.

1. Regarding hydrologic analyses, as stated above the GEV approach is appropriate, but describing results as percent changes in return period is simply not done in the field. The use of design return period changes is awkward at best and not the typical way to think about how precipitation changes will change infrastructure performance. Nowhere in practice are pdfs of return periods used (Figure 2). One might describe a shift in return period from say 10-yrs to 8.3-yrs, but that is not typically helpful in practice. The amplification factor approach is how this analysis should have been conducted throughout the paper. In the U.S. this would be called a safety factor. Section 2.1 should be removed. Section 2.2 should be in terms of required safety factors, not return period.

2. It is not clear what infrastructure means throughout the document. While it is neatly defined in the supplement, there needs to be a clear statement early on because the results cannot be interpreted without understanding which infrastructure is being impacted. The approach used as described in the supplement is reasonable – different assets have different design return periods and different parts of the globe might be expected to use a different return period. The methods and text never clearly identify which specific infrastructure is being analyzed nor how multiple infrastructure with differing return periods are combined. Developing nations appear to be analyzed in multiple manners. It would appear that analyzing each of these separately is more appropriate.

It might be helpful to be a bit more specific. Section 2.1 evaluates the future frequency of a 10-yr return period event – from Table S1 this return period is relevant to different assets by income group. Section 2.2 seems to combine all income groups and asset classes together in a manner that is unclear despite differences in return periods. This is probably why a percent change in return period was necessary. It seems that a separate reporting by asset class and insights to the diminished return periods by income would be a welcome addition.

3. The amount of effort that the authors put into the analysis is appreciated. Table S2, which provides the year that the GMT exceed 1.5 and 2.0oC by model, is very helpful. What is less clear is how the different models were combined to use a single value for each grid cell. The choice of using mid-century or late century periods vs GMT exceed 1.5 and 2.0oC is an important distinction and has value for very different audiences. It seems that the authors wish to reach the infrastructure community. If this is the case, then mid-century or late century periods would have been the appropriate choice. Otherwise, the work should be recrafted as a policy relevant document rather than providing guidance to the infrastructure design community.

Other comments

The opening is a litany of extreme weather events that are not related to climate change;

with paragraph two leading to concluding thoughts that are very loosely supported.

Extreme precipitation and anticipated changes due to climate change have an enormous literature. Vulnerability assessments abound. The authors failed to acknowledge the leading voices in that field. See a few references at the end of the review.

Lines 52 to 70 present results before the study is presented. Line 99 to 100 provide study goals in the middle of results.

A 1 in 10 event is rarely used as a design event for transportation infrastructure. Something seems wrong in lines 81 and 82 – there is an decrease for 1.5 oC and a very modest increase for 2oC.

Lines 84, 107 – do not use significant if there is not a statistical test.

Infrastructure and in particular drainage systems are almost always oversized due to “rounding up” to larger nominal pipe sizes. Failure of these systems does not mean damage except in very few cases – failure would be a bit of water backing up.

A better way of looking at this is the risk of the asset over its useful lifetime or design life (pick one)

Absolute exposure is likely a measure of density; relative exposure should be binary;

Line 138 Please define relative exposure and exposure ratio when it is used. This section has no meaning because these terms are not defined. It appears that Figure 4 is identical to figure

Line 155 – 157 methods in the middle of results where these methods are not clear on the approach or whether the median results approach is applied to all analyses or just the one that follows.

Lines 169-171 The statement “The smaller relative exposure changes in low and lower middle income countries are due to the lower design standards. The assets in these countries were already severely exposed to precipitation events with low return periods.” Appears to introduce an entirely new concept about design standards and whether assets are currently vulnerable.

Terms like high agreement are not defined (used in the paragraph starting on line 185)

Figure 5 does not make sense.

In section 2.3, “the ratio of the future precipitation intensity corresponding to the design return period of various transportation assets to the precipitation intensity in the baseline period.” Is finally used. This is something that engineers understand. And again in lines 247-248 “This indicates that the dynamic amplification factor is more robust in guiding engineering design compared to return period change ratio.” Yes! One looks at the change in design precipitation not changes in return period.

Line 248 – 249 “Considering the uncertainty in the tail of the PDFs of return period, using an amplification factor to current design precipitation intensity would be more feasible and easier.” There is easily a 10 to 20% uncertainty in the tails of the pdfs of the design rainfall.

The use of 1.5oC and 2.0oC rather than mid-century or late is a climate approach that is not

helpful in transportation design; the useful life of an asset dictates the design value for future precipitation. Thus, one must examine the range of precipitation values for a time period, not for a change in global mean temperature. Furthermore, the timing of temperature increases differs regionally making the use of a single GMT increase result in a the period of interest differing by region.

Precipitation intensity is a rate (mm/hr) not a depth (mm)

The conclusion loops back to drainage systems but the relationship between drainage systems and the performance of many km of railways and roads was never made. The concluding comment is unfounded "our results help to identify highest risk assets and facilitate decision making about which and where assets to prioritize upgrading drainage system, based on the potential increase in future exposure."

The recommendation of engineered solutions for updating drainage network is not well thought out. Given the cost of hard infrastructure and the relatively modest increases in precipitation intensity, the use of nature-based solutions and green infrastructure is an obvious, easy solution that should have been discussed.

Methods

RX1D is not an internationally recognized term.

What is the grid cell size? Smoothing precipitation parameters will grossly change the results and not account for local variations even at the GCM scale.

Where do the design return periods come from? The higher and lower values are not noted in the body of the analysis.

Estimation of exposure. It seems that relative exposure should be binary for each grid. Either all the roads are exposed or none; all the railways or none, etc. If this is the case then the relative exposure is effectively the ratio of the various types of assets in a grid.

Equation 3 – amplification is an average of the ratios of the different return period changes. Not a proper way of doing this analysis.

The literature on changing precipitation extremes is entirely missing. A few examples follow.

Trenberth, K.E., 2011. Changes in precipitation with climate change. *Climate research*, 47(1-2), pp.123-138.

O’Gorman, P.A., 2015. Precipitation extremes under climate change. *Current climate change reports*, 1(2), pp.49-59.

Wright, D.B., Bosma, C.D. and Lopez-Cantu, T., 2019. US hydrologic design standards insufficient due to large increases in frequency of rainfall extremes. *Geophysical Research Letters*, 46(14), pp.8144-8153.

Arnbjerg-Nielsen, K., Willems, P., Olsson, J., Beecham, S., Pathirana, A., Bülow Gregersen, I., Madsen, H. and Nguyen, V.T.V., 2013. Impacts of climate change on rainfall extremes and urban drainage systems: a review. *Water science and technology*, 68(1), pp.16-28.

Fowler, H.J., Ali, H., Allan, R.P., Ban, N., Barbero, R., Berg, P., Blenkinsop, S., Cabi, N.S., Chan, S., Dale, M. and Dunn, R.J., 2021. Towards advancing scientific knowledge of climate change impacts on short-duration rainfall extremes. *Philosophical Transactions of the Royal Society A*, 379(2195), p.20190542.

Lenderink, G. and Fowler, H.J., 2017. Understanding rainfall extremes. *Nature Climate Change*, 7(6), pp.391-393.

Fowler, H.J., Wasko, C. and Prein, A.F., 2021. Intensification of short-duration rainfall extremes and implications for flood risk: Current state of the art and future directions. Philosophical Transactions of the Royal Society A, 379(2195), p.20190541.

Global transportation infrastructure exposure to the change of precipitation in a warmer world

Manuscript ID: NCOMMS-22-02358A

We would like to thank the editor and reviewers for the thorough reading of the manuscript and the valuable remarks that helped us to improve the manuscript. We have revised the manuscript carefully according to the reviewers' comments, and have incorporated the suggestions into the revised manuscript. After having integrated these comments and suggestions, we believe it is now a vastly improved paper.

The notes below provide a point-by-point response to each comment from the referees. The texts with blue font are the reviewer's original comments, the texts with black font are authors' responses. If there is any question addressed unclearly or unsatisfied, we are always willing to make a second revision based on reviewer's comments. Thank you again for the opportunity to be considered for publication in *Nature Communications*.

Reviewer #1

General comments:

A global perspective on transport infrastructure design life and how these need to be rethought in the face of climate change – particularly, increasing precipitation projections – was enjoyable to read and think about. The paper is generally clearly written, with the authors providing simple and effective explanations of terms and concepts. However, I recommend some further editorial attention to sentence structure and grammar as some sentences were difficult to understand.

Response: We appreciate very much for reviewer's interest in the paper! In the revised manuscript, we have worked on both language and readability and have also involved professional language services. The language and sentence structures have been improved and we will be happy to edit the text further, based on the comments from the reviewer.

Specific comments:

I can only comment on the transport infrastructure aspects of the paper and not the climate science as the latter is out of my field of expertise. To this end I do have some questions and comments for the authors.

1. I would encourage authors to provide more details about how the transportation mapping work was done for this research, in order to facilitate reproduction. For example, were all the

roads (as per the classification from openstreetmaps) in urban regions represented? Also, I believe the analysis was done by assessing roadways within grid cells but more detail could be provided – why was 25x25km chosen, and how is the use of one size appropriate for both highly dense urbanized areas versus remote regions? What GIS software or coding platform was used to do the analysis? Related to this is the computational cost of running the analysis for the entire world at this gridding size.

Response: We thank for reviewer's comment and apologize for the unclear description. Following please find our specific response to each question.

1) **Regarding OSM data.** OpenStreet Map (OSM) is an open and free database of global geographic road, which is collected, edited and improved by a large number of volunteers. In particular, for roads and railways, OSM is now at a high level of completion and is considered to be the most consistent and complete global database of its type and has been widely used in various studies. Based on the OSM classification of roads, we chose the five most important categories of roads, i.e., (1) motorways, (2) trunks, (3) primary roads, (4) secondary roads, and (5) tertiary roads. Following the OSM definitions, these categories are interpreted in the following manner: (1) A motorway is a restricted access major divided highway, normally with two or more running lanes plus an emergency hard shoulder. (2) Trunks are the most important roads in a country's system that are not motorways. Such roads need not be divided highways. (3) Primary roads are the next most important roads in a country's system and often link larger towns. (4) Secondary roads are the next most important roads and often link towns. (5) Tertiary roads are the next most important roads and often link smaller towns and villages. Some small roads are not included, such as footways, because it is often uncertain whether a drainage design was considered when such footways were constructed. Additionally, the largest gaps in the OSM data are primarily in the lower tier roads, including footways and residential roads. For railways, we selected rails with full-sized passenger or freight trains in the standard gauge for the given country or state.

The road and rail data used in this study were downloaded from the free Geofabrik download server, which is usually updated daily. Our data were updated as of May 6, 2021. The data consist of linear vector data containing spatial positions. Table S3 gives a detailed description and the length of the different categories of transportation assets, and their spatial distribution is given in Supplementary Figure S2.

In the revised manuscript, we have added the above information in the Data section in pages 20-21.

2) **Regarding grid size.** The analysis grid cell of ~25 km×25 km was chosen according to the resolution of climate models. In this study, we use the NASA Earth Exchange Global Daily Downscaled Projections (NEX-GDDP) dataset. It is comprised of downscaled climate

scenarios for the globe that are derived from the General Circulation Model (GCM) runs conducted under the Coupled Model Intercomparison Project Phase 5 (CMIP5) and across two of the four greenhouse gas emissions scenarios known as Representative Concentration Pathways (RCPs). The spatial resolution of the dataset is 0.25 degrees (~25 km×25 km), which is the best data we can get at the global scale. Furthermore, although we present the results by grids, the transportation infrastructure is vector data and the length of transportation infrastructure in each grid is counted. In the revised manuscript, we have added the description of NEX-GDDP in the Data section in page 20 and also added the following description in the Method section in lines 364-366, page 18:

A grid size of approximately 25 km × 25 km was chosen according to the resolution of the climate models. The transportation infrastructure consists of vector data, and its length in each grid was counted.

- 3) **Regarding Calculation.** We used python programming language to perform the analysis. There are a total of 720×1440 grids with each grid size of ~25 km×25 km. The computational cost is ~20 minutes with AMD Ryzen 7 3700X 8-Core Processor (3.59 GHz 32G RAM) for one climate model without considering data preprocessing and post processing. We have added code availability in the revised manuscript in lines 431-433, page 21:

Code availability

We use python programming language to perform the analysis. Codes are available through the following DOI: <https://github.com/Vapson/InfExposure>.

2. The analysis and results entirely depend upon the (current) design periods assumed for each of the three country categories (low, upper middle and high income) and their 3 categories of infrastructure. Although highly coarse, given the global scope of the analysis I can appreciate why this was adopted. However, infrastructure design standards are often particular to jurisdictions (i.e., metropolitan regions, counties, states) in many countries, each with their particular populations, infrastructure characteristics and climates. Thus, is design life relatively consistent throughout a country? Between countries in a region of the world? Given the heavy reliance of the results on assumed design life, I believe some background and discussion on design life and variations throughout the world would be very helpful to have better perspective on the results, and to understand their limitations with respect to applicability and interpretation. In addition, who is the intended audience for this paper (asides academic researchers) – would it be an international organization like the World Bank? Or individual nations? Or, say, individual municipalities that must make changes to design life? I don't believe this analysis has relevance to the latter, and thus I believe it would be nice to have some discussion of this point. On a related note, on page 5, referring to design return period

“historical period 1971–2000,” what exactly does this mean? Would this be the assumed return period for each income group country for each type of asset?

Response: We appreciate the reviewer’s understanding on our assumption of the design standards in different countries considering this is a global scale analysis. Our specific responses to each question are given below:

- 1) The design standards are generally consistent throughout a single country, however, for some vast countries, such as the United States, the consistency in design life could be achieved at the state or federal level. Also how well these infrastructures are maintained and the deterioration rates are different across the world and across the country. These factors are not considered in this study given the global scale of the analysis, and the lack of availability of such data. We suggest that more detailed or regional analysis using local data can be performed followed our framework to consider regional difference within the country based on our shared code. In this revised manuscript, we have added discussion on design life and acknowledged this limitation in the Method part in lines 333-340, page 17:

However, we admit that large uncertainty exists in the design standards across the world. Additionally, even though design standards are generally relatively consistent throughout a single country, for some vast countries, such as the United States, the consistency in the design life can be determined at the state or federal level. Furthermore, how well infrastructure is maintained and its deterioration rate differ across the world and across a given country. These factors were not considered in this study given the global scale of the analysis and the lack of data availability. Therefore, this study should be considered as a starting point for the identification of global hotspots, from which more detailed analyses can be performed using local data.

- 2) We believe that Nature Communications is the right place to help convey our insights to a broader audience. The intended audience for this paper include three groups. **Firstly**, the academic researchers. The attention given to infrastructure adaptation to climate change is relatively recent but its importance is likely to rise. Our study improves understanding of climate impacts on transportation assets. **Secondly**, International organizations like the World Bank, the Asian Development Bank, etc. Our findings underscore the differences in the impact of climate change in transportation assets of different countries and provide a global view of hotspots where risks are the highest. This information can guide decisions regarding priority regions for adaptation to more extreme rainfall, as it is too expensive to increase design standards everywhere, the funds should be spent more efficiently and more optimal. When moving towards implementation of individual measures, detailed analysis should be performed using local models and data. **Thirdly**, Local Governments and Municipalities. Understanding the impact of climate change on infrastructure assets is crucial to design adaptation strategies. Our study provides a picture of future risk faced by

transportation infrastructure, and helps to bridge the knowledge gaps among climate science communities, infrastructure communities and policy makers. This knowledge can also incentivize the local governments or municipalities to design measures to cope with the risk faced by infrastructures with anthropogenic warming.

In the revised manuscript, we have followed reviewers' suggestions to add discussion of this point in the Discussion and Conclusions Section in lines 283-298, page 15:

To conclude, we emphasize that it is essential to acknowledge uncertainties when designing resilient infrastructure for the future. In particular, in the case of infrastructure, which generally has a long lifespan, it is essential to incorporate potential future changes in the exposure to maintain the reliability of the infrastructure over its multi-decadal service life span given growing climatic uncertainty. Our findings underscore the differences in the impacts of climate change on transportation assets in different countries and provide a global view of hotspots at high risk. Such information is essential in facilitating decision-making regarding the prioritization of regions for adaptation to more extreme rainfall because it is too expensive to increase drainage design standards everywhere. Identifying the regions most at risk will allow for more efficient and optimal spending of the available funds. When moving toward the implementation of individual measures, detailed analyses need to be performed using local models and data. Furthermore, understanding the impact of climate change on infrastructure assets is crucial to design adaptation strategies for local governments and municipalities. Our study provides a picture of future risks faced by transportation infrastructure and helps bridge the knowledge gap between climate science communities, infrastructure communities, and policy makers. This knowledge can also incentivize local governments or municipalities to design infrastructure adaptation measures to cope with the risk of anthropogenic warming. Climate researchers, construction designers, and stakeholders need to work together to make infrastructure reliable in a warming world.

- 3) To evaluate the change of precipitation in the future, we have to define a baseline time period for comparison. Since the design information for drainage system of transportation infrastructure is not available, we assume they were designed or updated based on the precipitation record of 1971-2000 in this study. More detailed analysis can be performed based on more local data. In the revised manuscript, we have added this information in lines 304-307, page 16:

Because detailed design information for transportation drainage systems is not available for each country, we assumed that these systems were designed or updated based on the precipitation record for the period 1971–2000 used in this study, i.e., the baseline period.

3. The authors do point out that the results of absolute exposure metric are influenced by both network density and change in precipitation design return period. It seems this metric could be much more useful if the authors could provide some benchmark (map?) of network density, because as it stands it looks to be far more a reflection of network density rather than change in return period.

Response: We thank for reviewer's comment. As suggested, we have added the transportation spatial distribution map in supplementary file as can be found in Figure S2. Furthermore, in the revised version, we have also provided a separate reporting by asset class and insights to the diminished return periods in the supplementary file (Figures S8-13).

4. I believe the assertion in the first line of 2nd paragraph, p 15 should be revised. The paper is illuminating about where in the world assets may be at greatest risk, but standards on how to prioritize drainage system upgrades seems a specifically jurisdictional level issue that I don't believe this work is able to address (they are at differing scales). This should be a relatively simple editorial fix.

Response: We agree with the reviewer's comment. Indeed our study aims to help the national or international body to highlight the hotspots of transportation infrastructure, certain and detailed research can be done on the more local level to identify assets at high risk. More research at local scales is needed so as to include the information into design guidelines. In the revised manuscript, we have almost rewrote the Discussion and Conclusions section and please also refer to our response to Question 2.

Reviewer #2

The study's strength is that it takes a global look at infrastructure using open street map to extract roads and railways and a downscaled 25 km climate product that is translated using the GEV method to design precipitation and combines. So nice datasets, neatly set up. The manuscript is well written with just a need for a minor grammar edit and the figures are appropriate with a need for slightly more descriptive captions.

Response: Thank you very much for the interest on the topic and data and helpful comments. As suggested, we have worked on the language and have also involved professional language services. We also add more descriptive captions for figures. We will be happy to edit the text further, based on the comments from the reviewer.

1. The main problems stem from a lack of understanding regarding hydrologic and hydraulic engineering as well as that role in transportation. Regarding hydrologic analyses, as stated

above the GEV approach is appropriate, but describing results as percent changes in return period is simply not done in the field. The use of design return period changes is awkward at best and not the typical way to think about how precipitation changes will change infrastructure performance. Nowhere in practice are pdfs of return periods used (Figure 2). One might describe a shift in return period from say 10-yrs to 8.3-yrs, but that is not typically helpful in practice. The amplification factor approach is how this analysis should have been conducted throughout the paper. In the U.S. this would be called a safety factor. Section 2.1 should be removed. Section 2.2 should be in terms of required safety factors, not return period.

Response: We thank for editor's comment. Our specific responses to each question are as follows:

1) Drainage system protects road and railway from water and is an important consideration in the design. In practice pdfs of return periods, i.e., the frequency at which the corresponding precipitation can be expected to occur, are usually used for designing drainage facilities. Return period gives the probability that events will occur. If there is a 50% decreases in return period, it means the likelihood that such event occurs has doubled. Since drainage systems are designed based on certain return period, we think it is a good indicator to show the change in the potential hazard. It helps people to have a better understanding on how frequent a certain event may occur, which is particularly useful for International organizations like the World Bank, the Asian Development Bank, etc. In the next step in 2.3, we calculated the 'safety factor' to guide infrastructure design. Both return period change and precipitation amount change are good indices to characterize future change. Hence, we would like to keep section 2.1. To avoid the confusion, we have added more descriptive explanations in the revised manuscript in section 2.1, as can be found in lines 79-87, page 4:

The changes in the precipitation return periods between the present (1971–2000) and two future horizons (2030–2059 and 2070–2099) were analyzed. The magnitudes of the precipitation for certain return periods (for example, 2, 5, 10, 20, 30, and 50 years) in the present day were calculated for each grid. The corresponding return period of the same magnitude precipitation was then computed for the time series of the future precipitation for each grid under different climate scenarios. The details of the above processes are described in the Methods.

Figure 1 shows an example of how the probability of a 1-in-10-year precipitation event (an event that has, on average, a 10% chance of occurring each year) may change in the future. The results for other return periods (Supplementary Figures S3–S7) show similar spatial distributions, even though the magnitude varies.

2) We agree with the reviewer that 'safety factor' is more appropriate than 'amplification factor'. In engineering practice, it is common to employ a safety factor. If a safety factor of 2 is employed when building a bridge, then the bridge is calculated to resist twice the maximal

load for which is intended (Möller et al.). To emphasize that our proposed safety factor is additionally introduced for coping with climate change, we have named it ‘safety factor for climate change adaptation’, which makes this concept more clearer.

Ref: Niklas Möller, Sven Ove Hansson. Principles of engineering safety: Risk and uncertainty reduction. Reliability Engineering & System Safety. 93:798-805.2008.

2. It is not clear what infrastructure means throughout the document. While it is neatly defined in the supplement, there needs to be a clear statement early on because the results cannot be interpreted without understanding which infrastructure is being impacted. The approach used as described in the supplement is reasonable – different assets have different design return periods and different parts of the globe might be expected to use a different return period. The methods and text never clearly identify which specific infrastructure is being analyzed nor how multiple infrastructure with differing return periods are combined. Developing nations appear to be analyzed in multiple manners. It would appear that analyzing each of these separately is more appropriate. It might be helpful to be a bit more specific. Section 2.1 evaluates the future frequency of a 10-yr return period event – from Table S1 this return period is relevant to different assets by income group. Section 2.2 seems to combine all income groups and asset classes together in a manner that is unclear despite differences in return periods. This is probably why a percent change in return period was necessary. It seems that a separate reporting by asset class and insights to the diminished return periods by income would be a welcome addition.

Response: We thank for reviewer’s helpful comments. Following please find our specific responses to each question.

- 1) We agree with the reviewer that it would be clearer to give a description of the infrastructure analyzed at the beginning. In the revised manuscript, we have added a general description in of the infrastructure in the introduction section at the beginning for a better interpretation in lines 52-54, page 3:

Here, we analyze future changes in the global exposure of roads and railways (see Data and Table S1) to precipitation in a warmer world using multi-model projections from the Coupled Model Intercomparison Project Phase 5 (CMIP5).

- 2) More detailed explanation of OSM is provided in the lines 409-428, page 20-21, and an additional figure to show the spatial distribution of infrastructure is added in Supplementary Figure 2.

OpenStreet Map (OSM) is an open and free database of global geographic road, which is collected, edited and improved by a large number of volunteers. In particular, for roads and railways, OSM is now at a high level of completion and is considered to be the most

consistent and complete global database of its type and has been widely used in various studies. Based on the OSM classification of roads, we chose the five most important categories of roads, i.e., (1) motorways, (2) trunks, (3) primary roads, (4) secondary roads, and (5) tertiary roads. Following the OSM definitions, these categories are interpreted in the following manner: (1) A motorway is a restricted access major divided highway, normally with two or more running lanes plus an emergency hard shoulder. (2) Trunks are the most important roads in a country's system that are not motorways. Such roads need not be divided highways. (3) Primary roads are the next most important roads in a country's system and often link larger towns. (4) Secondary roads are the next most important roads and often link towns. (5) Tertiary roads are the next most important roads and often link smaller towns and villages. Some small roads are not included, such as footways, because it is often uncertain whether a drainage design was considered when such footways were constructed. Additionally, the largest gaps in the OSM data are primarily in the lower tier roads, including footways and residential roads. For railways, we selected rails with full-sized passenger or freight trains in the standard gauge for the given country or state.

The road and rail data used in this study were downloaded from the free Geofabrik download server, which is usually updated daily. Our data were updated as of May 6, 2021. The data consist of linear vector data containing spatial positions. Table S3 gives a detailed description and the length of the different categories of transportation assets, and their spatial distribution is given in Supplementary Figure S2.

- 3) We agree with the reviewer that each type of assets should be analyzed separately, which is also the approach we used. However, due to the size restriction, we are unable to present separate result by asset class in the main text. According to reviewer's suggestion, we have provided a separate reporting by asset class and return periods change in supplementary Figs. S8-S13. Furthermore, to make it clear, we have revised the description about the method on how multiple infrastructure with differing return periods are combined in the Method section in lines 366-380:

We assumed that, when there was a more than 25% decrease in the design return period, the drainage system would be inadequate to protect the transportation infrastructure. The absolute exposure (AE) is defined as the total number of transportation assets exposed to a more than 25% decrease in the design return period in the future. We calculated the change in the return period for each type of asset separately and then added the exposed length of the different types of assets to obtain the absolute exposure. The relative exposure (RE) is defined as the ratio of the absolute exposure to the total assets within a grid. AE and RE can be calculated using Eqs. (1) and (2), respectively.

$$AE_g = \sum_i^n EL_{i,g} \quad (1)$$

$$RE_g = \frac{AE_g}{TL_g} = \frac{1}{TL_g} \times \sum_i^n EL_{i,g} \quad (2)$$

Here, AE_g is the absolute exposure of the grid g ; RE_g is the relative exposure of the grid g ; i is the type of asset, with a total of n types; EL_g is the length of the assets exposed to a 25% decrease in the design return period in the future within the grid g ; and TL_g is the total length of the assets within the grid g .

- 4) We calculated the change in return period for different time periods (for example, 2, 5, 10, 20, 30, 50 years) for each grid. The future frequency of a 10-yr return period event is presented as an example in Section 2.1. To avoid confusion, we have added the following explanation in lines 79-87, page 4 in section 2.1 and also add the results for other return periods in Supplementary Figs.S3- S7:

The changes in the precipitation return periods between the present (1971–2000) and two future horizons (2030–2059 and 2070–2099) were analyzed. The magnitudes of the precipitation for certain return periods (for example, 2, 5, 10, 20, 30, and 50 years) in the present day were calculated for each grid. The corresponding return period of the same magnitude precipitation was then computed for the time series of the future precipitation for each grid under different climate scenarios. The details of the above processes are described in the Methods.

Figure 1 shows an example of how the probability of a 1-in-10-year precipitation event (an event that has, on average, a 10% chance of occurring each year) may change in the future. The results for other return periods (Supplementary Figures S3–S7) show similar spatial distributions, even though the magnitude varies.

3. The amount of effort that the authors put into the analysis is appreciated. Table S2, which provides the year that the GMT exceed 1.5 and 2.0°C by model, is very helpful. What is less clear is how the different models were combined to use a single value for each grid cell. The choice of using mid-century or late century periods vs GMT exceed 1.5 and 2.0°C is an important distinction and has value for very different audiences. It seems that the authors wish to reach the infrastructure community. If this is the case, then mid-century or late century periods would have been the appropriate choice. Otherwise, the work should be recrafted as a policy relevant document rather than providing guidance to the infrastructure design community.

Response: We thank for reviewer's comment. Following please find our specific responses to each question:

- 1) For each grid cell, we present the median value of different climate models. We have revised both the text and the caption of the figure to make it more clearer.
- 2) We agree with the reviewer that it would be more appropriate to present the results for mid-century and late-century. In the revised manuscript, we have made new calculations and changed the results of different warming levels to different time periods, as suggested by the reviewer. Please refer to the results presented in the revised manuscript.

Other comments:

1. The opening is a litany of extreme weather events that are not related to climate change; with paragraph two leading to concluding thoughts that are very loosely supported.

Response: We thank for editor's comment. In the revised manuscript, we have revised the opening to make it more readable.

2. Extreme precipitation and anticipated changes due to climate change have an enormous literature. Vulnerability assessments abound. The authors failed to acknowledge the leading voices in that field. See a few references at the end of the review.

Response: We thank for editor's comment. We have added references related to extreme precipitation and climate change and revised the Introduction section. Specifically, the following references are added:

Trenberth, K.E., 2011. Changes in precipitation with climate change. Climate research, 47(1-2), pp.123-138.

O'Gorman, P.A., 2015. Precipitation extremes under climate change. Current climate change reports, 1(2), pp.49-59.

Wright, D.B., Bosma, C.D. and Lopez-Cantu, T., 2019. US hydrologic design standards insufficient due to large increases in frequency of rainfall extremes. Geophysical Research Letters, 46(14), pp.8144-8153.

Arnbjerg-Nielsen, K., Willems, P., Olsson, J., Beecham, S., Pathirana, A., Bülow Gregersen, I., Madsen, H. and Nguyen, V.T.V., 2013. Impacts of climate change on rainfall extremes and urban drainage systems: a review. Water science and technology, 68(1), pp.16-28.

Fowler, H.J., Ali, H., Allan, R.P., Ban, N., Barbero, R., Berg, P., Blenkinsop, S., Cabi, N.S., Chan, S., Dale, M. and Dunn, R.J., 2021. Towards advancing scientific knowledge of climate change impacts on short-duration rainfall extremes. Philosophical Transactions of the Royal Society A, 379(2195), p.20190542.

Lenderink, G. and Fowler, H.J., 2017. Understanding rainfall extremes. *Nature Climate Change*, 7(6), pp.391-393.

Fowler, H.J., Wasko, C. and Prein, A.F., 2021. Intensification of short-duration rainfall extremes and implications for flood risk: Current state of the art and future directions. *Philosophical Transactions of the Royal Society A*, 379(2195), p.20190541.

Seneviratne S.I. et al. , 2021. *Weather and Climate Extreme Events in a Changing Climate*.

Easterling, D. R. et al., 2000. Climate extremes: observations, modeling, and impacts. *science* 289, 2068-2074.

Fischer, E. M., Beyerle, U. & Knutti, R. , 2013. Robust spatially aggregated projections of climate extremes. *Nature Climate Change* 3, 1033-1038.

Westra, S. et al., 2014. Future changes to the intensity and frequency of short-duration extreme rainfall. *Reviews of Geophysics* 52, 522-555.

Westra, S., Alexander, L. V. & Zwiers, F. W., 2013. Global increasing trends in annual maximum daily precipitation. *Journal of climate* 26, 3904-3918.

3. Lines 52 to 70 present results before the study is presented. Line 99 to 100 provide study goals in the middle of results.

Response: We thank for reviewer's comment. In the revised manuscript, we have removed lines 91 to 92 into the introduction section before the results is presented, which can be seen in lines 52-54, page 3:

Here, we analyze future changes in the global exposure of roads and railways (see Data and Table S1) to precipitation in a warmer world using multi-model projections from the Coupled Model Intercomparison Project Phase 5 (CMIP5).

4. A 1 in 10 event is rarely used as a design event for transportation infrastructure. Something seems wrong in lines 81 and 82 – there is an decrease for 1.5 °C and a very modest increase for 2°C.

Response: We thank for reviewer's comment. Indeed the design standard varies with country. For example, in New York, the design frequency for drainage system is 10 years for Freeways, and 5 years for local roads. In the UK, the design frequency is 10 years for highways. In China, the design frequency is 5 years for highways and first-class roads, and 3 years for other roads. While in the case of Nepal, the recommended design year of rainfall probability is 3 years for highways, 2 years for local roads and 1 year for other roads. Based on the collected data, we make an assumption and assign design return periods of drainage system to each country according to its income level, as given in Table S1. We have calculated changes in different

return periods and the result of 1 in 10 year is given as an example in the main text. We have added the following information in lines 79-87, page 4 to make it clear:

The changes in the precipitation return periods between the present (1971–2000) and two future horizons (2030–2059 and 2070–2099) were analyzed. The magnitudes of the precipitation for certain return periods (for example, 2, 5, 10, 20, 30, and 50 years) in the present day were calculated for each grid. The corresponding return period of the same magnitude precipitation was then computed for the time series of the future precipitation for each grid under different climate scenarios. The details of the above processes are described in the Methods.

Figure 1 shows an example of how the probability of a 1-in-10-year precipitation event (an event that has, on average, a 10% chance of occurring each year) may change in the future. The results for other return periods (Supplementary Figures S3–S7) show similar spatial distributions, even though the magnitude varies.

We also admit that uncertainty existed in our analysis in lines 333-340, page 17:

However, we admit that large uncertainty exists in the design standards across the world. Additionally, even though design standards are generally relatively consistent throughout a single country, for some vast countries, such as the United States, the consistency in the design life can be determined at the state or federal level. Furthermore, how well infrastructure is maintained and its deterioration rate differ across the world and across a given country. These factors were not considered in this study given the global scale of the analysis and the lack of data availability. Therefore, this study should be considered as a starting point for the identification of global hotspots, from which more detailed analyses can be performed using local data.

5. Lines 84, 107 – do not use significant if there is not a statistical test.

Response: We agree with reviewer’s comment. In the revised manuscript, we have deleted the term of ‘significant change’.

6. Infrastructure and in particular drainage systems are almost always overdesigned due to “rounding up” to larger nominal pipe sizes. Failure of these systems does not mean damage except in very few cases – failure would be a bit of water backing up.

Response: We thank for reviewer’s comment. In this study, we focus on the transportation infrastructure exposure due to inadequate designed drainage. When the precipitation exceeds the design threshold, the excess water (water backing up could be one of the reasons) will adversely affect the transport assets, such as cut or fill slope failures, road surface erosion, etc. While the damage of drainage pipe is not considered in this study. To avoid the confusion, we have added the following description in lines 107-108, page 5:

When the probability of a rainfall event exceeds this design threshold, the excess water will adversely affect the transport assets; such effects include cut or fill slope failures, road surface erosion, etc.

7. A better way of looking at this is the risk of the asset over its useful lifetime or design life (pick one)

Response: We thank for reviewer's comment. We agree with the reviewer that it is of great importance to evaluate the risk of assets over design life. In this study, we focus on which assets are hotspots exposed to future climate change, which is a starting point for risk analysis. To do so, more work should be done on the vulnerability of the transportation assets to extreme rainfall, which is still rare. We have added the following in the discussion in the revised manuscript in lines 275-276, page 14:

To assess whether it is cost effective to implement such measures, the risk of a transportation asset during its design life with respect to future climate change needs to be analyzed and a cost-benefit analysis is required.

8. Absolute exposure is likely a measure of density; relative exposure should be binary;

Response: We thank for reviewer's comment. The absolute exposure is determined by both the transportation density and the change of precipitation return period. Only the assets exposed to 'more frequent extreme precipitation' within a grid, expressed as a 25% decrease in the design return period in the future, is counted. As for the relative exposure, it is indeed binary if there is only one type of assets. However, since we have different types of assets, the relative exposure in one grid is expressed as the ratio of absolute exposure to the total assets' length. Absolute exposure and Relative exposure are calculated according to eq.1 and eq.2:

$$AE_g = \sum_i^n EL_{i,g} \quad (1)$$

$$RE_g = \frac{AE_g}{TL_g} = \frac{1}{TL_g} \times \sum_i^n EL_{i,g} \quad (2)$$

Here, AE_g is the absolute exposure of the grid g ; RE_g is the relative exposure of the grid g ; i is the type of asset, with a total of n types; EL_g is the length of the assets exposed to a 25% decrease in the design return period in the future within the grid g ; and TL_g is the total length of the assets within the grid g .

According to Eq. (2), if we assume that there are 5 km of primary roads and 15 km of tertiary roads in a grid, if the primary road experiences more frequent extreme precipitation, i.e., a 25% decrease in the design return period in the future, while the secondary roads do not, then the absolute exposure is 5 km and the relative exposure is $5 \text{ km} / (5 + 15) \text{ km} = 25\%$.

To make it more clearer, we have added the above example in the method section in lines 382-385, page 19.

9. Line 138 Please define relative exposure and exposure ratio when it is used. This section has no meaning because these terms are not defined. It appears that Figure 4 is identical to figure ?

Response: We apologize for the unclear definition. In the revised manuscript, we have added the definition of relative and exposure ratio in lines 154-155, page 8 to make it clear:

Figure 4 shows the relative global transport infrastructure exposure; that is, the ratio of the absolute exposure to the total assets (see Methods).

10. Line 155 – 157 methods in the middle of results where these methods are not clear on the approach or whether the median results approach is applied to all analyses or just the one that follows.

Response: We apologize for the unclear description. In the revised manuscript, we only keep the median results and emphasized it in both the description and the caption of related figures.

11. Lines 169-171 The statement “The smaller relative exposure changes in low and lower middle income countries are due to the lower design standards. The assets in these countries were already severely exposed to precipitation events with low return periods.” Appears to introduce an entirely new concept about design standards and whether assets are currently vulnerable.

Response: We thank for reviewer’s comment. Lines 161-163 is related to Figure 5 in the previous submission. In the revised manuscript, we have followed reviewer’s comment (Question 13) to remove Figure 5 in the revised manuscript, the corresponding descriptions are deleted as well.

12. Terms like high agreement are not defined (used in the paragraph starting on line 185)

Response: We thank for editor’s comment. In the revised manuscript, we only keep the median result.

13. Figure 5 does not make sense.

Response: We thank for reviewer's comment. We have deleted Figure 5 and corresponding descriptions as suggested by the reviewer.

14. In section 2.3, "the ratio of the future precipitation intensity corresponding to the design return period of various transportation assets to the precipitation intensity in the baseline period." is finally used. This is something that engineers understand. And again in lines 247-248 "This indicates that the dynamic amplification factor is more robust in guiding engineering design compared to return period change ratio." Yes! One looks at the change in design precipitation not changes in return period.

Response: We thank for reviewer's comment. We agree with the reviewer that the amplification factor (safety factor in the revised manuscript) is important. Throughout the paper, we are working to emphasize the importance of safety factor step by step. As we explained in question 1, the change in return period helps people to have a better understanding on how frequent of certain event will occur, this is particularly useful for International organizations like the World Bank, the Asian Development Bank, etc. In the next step in 2.3, we calculated the safety factor to guide infrastructure design. The return period change and precipitation amount change are two aspects to characterize future change, and are useful for different audiences. To avoid confusion, we have deleted "This indicates that the amplification factor is more robust in guiding engineering design compared to return period change ratio" in the revised manuscript, as it is unnecessary to compare them.

15. Line 248 – 249 "Considering the uncertainty in the tail of the PDFs of return period, using an amplification factor to current design precipitation intensity would be more feasible and easier." There is easily a 10 to 20% uncertainty in the tails of the pdfs of the design rainfall.

Response: We thank for reviewer's comment. We totally agree with the reviewer. In the revised manuscript, we have deleted the comparison between change in return period and amplification factor.

16. The use of 1.5°C and 2.0°C rather than mid-century or late is a climate approach that is not helpful in transportation design; the useful life of an asset dictates the design value for future precipitation. Thus, one must examine the range of precipitation values for a time period, not for a change in global mean temperature. Furthermore, the timing of temperature increases differs regionally making the use of a single GMT increase result in a the period of interest differing by region.

Response: We thank for reviewer's comment. In the revised manuscript, we have followed reviewer's suggestion to change the use of temperature increase into different time period (mid-century and late century). All figures and corresponding descriptions have been updated.

17. Precipitation intensity is a rate (mm/hr) not a depth (mm)

Response: We thank for editor's comment. In this study, RX1D is defined as annual maximum daily precipitation, hence the unit is mm/day. We have revised the unit in the revised manuscript.

18. The conclusion loops back to drainage systems but the relationship between drainage systems and the performance of many km of railways and roads was never made. The concluding comment is unfounded "our results help to identify highest risk assets and facilitate decision making about which and where assets to prioritize upgrading drainage system, based on the potential increase in future exposure."

Response: We thank for reviewer's comment. In this study, we focus on the transportation infrastructure exposure due to inadequate designed drainage. Drainage system protects road and railway from water and is an important consideration in the design. When the precipitation exceeds the design threshold, the excess water (water backing up could be one of the reasons) will adversely affect the transport assets, such as cut or fill slope failures, road surface erosion, etc. In this study, we assumed that when there is a more than 25% decrease in the design return period, the drainage system will be inadequate to protect the transportation infrastructure. The Absolute exposure (AE) is therefore defined as the total amount of transportation assets exposed to a more than 25% decrease in the design return period. To avoid the confusion, we have added the following description in lines 366-370, page 18:

We assumed that, when there was a more than 25% decrease in the design return period, the drainage system would be inadequate to protect the transportation infrastructure. The absolute exposure (AE) is defined as the total number of transportation assets exposed to a more than 25% decrease in the design return period in the future. We calculated the change in the return period for each type of asset separately and then added the exposed length of the different types of assets to obtain the absolute exposure.

19. The recommendation of engineered solutions for updating drainage network is not well thought out. Given the cost of hard infrastructure and the relatively modest increases in precipitation intensity, the use of nature-based solutions and green infrastructure is an obvious, easy solution that should have been discussed.

Response: We appreciate for reviewer's suggestion. Indeed nature based solutions can offer a cost-effective way to the management of rainwater or surface water. We have added the following in the discussion in lines 278-281, page 15:

In addition to increasing design standards, natural solutions and green infrastructure can be relatively cost-effective strategies²⁵⁻²⁷ that can be integrated with existing drainage systems to reduce the impacts of excessive precipitation. Optimal risk reduction strategies could consist of a mixture of different measures.

Methods

20. RX1D is not an internationally recognized term.

Response: We apologize for the unclear definition. RX1D is defined as annual maximum daily precipitation and is one of indices to characterize extreme precipitation intensity. We have added this references for RX1D in the revised manuscript:

Min, SK., Zhang, X., Zwiers, F. et al. Human contribution to more-intense precipitation extremes. Nature 470, 378–381 (2011). <https://doi.org/10.1038/nature09763>

Pfahl, S., O’Gorman, P. & Fischer, E. Understanding the regional pattern of projected future changes in extreme precipitation. Nature Clim Change 7, 423–427 (2017). <https://doi.org/10.1038/nclimate3287>

Madakumbura, G.D., Thackeray, C.W., Norris, J. et al. Anthropogenic influence on extreme precipitation over global land areas seen in multiple observational datasets. Nat Commun 12, 3944 (2021). <https://doi.org/10.1038/s41467-021-24262-x>

21. What is the grid cell size? Smoothing precipitation parameters will grossly change the results and not account for local variations even at the GCM scale.

Response: We thank for reviewer’s comment. The grid cell size is 0.25 degrees (~25 km×25 km), which is the best climate data we can obtain at the global scale. We have added this information throughout the manuscript.

We agree with the reviewer’s comment that smoothing is will change the results. In the revised manuscript, we have updated all calculations without considering smoothing precipitation parameters.

22. Where do the design return periods come from? The higher and lower values are not noted in the body of the analysis.

Response: We thank for reviewer’s comment. The design standard varies with country. To perform this analysis, we have try to collect the design standards of different countries, although not too much can be found only. In New York, the design frequency for drainage system is 10 years for Freeways, and 5 years for local roads. In the UK , the design frequency is 10 years for highways. In China, the design frequency is 5 years for highway and first class road, and 3 years for other road. While in case of Nepal, the recommended design year of rainfall probability is 3 years for Highway, 2 years for local roads and 1 year for other roads. Generally, the design standard is highly related to the developing level, high income countries usually have a higher design standards. We therefore divide countries into different developing levels to account for their differences in design standards. Furthermore, as large uncertainties exist, we

set higher and lower values for different transportation assets. We admit that a large uncertainty exists, more detailed analysis can be performed based on more local data. In the revised manuscript, we have admitted this limitation in the Method section. Please also refer to our answer to question 4.

23. Estimation of exposure. It seems that relative exposure should be binary for each grid. Either all the roads are exposed or none; all the railways or none, etc. If this is the case then the relative exposure is effectively the ratio of the various types of assets in a grid.

Response: We thank the reviewer for pointing this issue. It is indeed binary if there is only one type of assets. The relative exposure (RE) as the ratio of the absolute exposure to the total assets within a grid. Please refer to our response to question 8.

24. Equation 3 – amplification is an average of the ratios of the different return period changes. Not a proper way of doing this analysis.

Response: We agree with the reviewer that it is more appropriate to present the amplification factor of each asset. However, as restricted by the manuscript space, we are unable to put too many figures in the main file, the separate results for different assets are put in Supplementary Figs.S16-S21 instead in the revised manuscript. Actually we found that there is little difference between the amplification factors (safety factor in the revised manuscript) of different assets, which are ranging from 1-1.5. It is reasonable to give the average average amplification factor of each grid to give a general picture for readers, we therefore still keep it in the main manuscript. To make it more clearly, we have revised equation 3 as:

$$SF_{g,i} = \frac{pre_{g,i}^f}{pre_{g,i}^p} \quad (3)$$

$$SF_g = \frac{1}{n} \times \sum_i^n SF_{g,i} \quad (4)$$

Here, $SF_{g,i}$ is the safety factor of the i th type of asset in the grid g ; $pre_{g,i}^f$ is the future precipitation corresponding to the design return period of the i th type of asset in the grid g ; $pre_{g,i}^p$ is the contemporary precipitation corresponding to the design return period of the i th type of asset in the grid g ; and n is the number of asset types.

REVIEWER COMMENTS

Reviewer #1 (Remarks to the Author):

The authors have done a fine job in addressing reviewer comments and improving this paper, and I do commend them for this effort! I have a couple follow up points that they might wish to address.

1. I am still struggling with the absolute exposure metric and what it tells us (and how you discuss the results). I completely understand its use. However, it simply appears to be dominated by infrastructure density (and country size) at the top (no surprises here), while as one moves down the list I guess extreme weather and other factors starts playing a part? I can only speculate as these dynamics and their nuances, and how relative exposure compares and relates, are not discussed. It would really help to discuss these results further and more deeply, to provide illumination.

2. The authors added: "Furthermore, understanding the impact of climate change on infrastructure assets is crucial to design adaptation strategies for local governments and municipalities. Our study provides a picture of future risks faced by transportation infrastructure and helps bridge the knowledge gap between climate science communities, infrastructure communities, and policy makers. This knowledge can also incentivize local governments or municipalities to design infrastructure adaptation measures to cope with the risk of anthropogenic warming." These statements are rather obvious and are not new or insightful observations. As far as I can see, this work does not offer tangible and direct guidance to local governments, who need more detailed and context-sensitive advice. I would suggest either removing any mention of impact to local authorities, or considering further exactly how this work directly impacts them.

Reviewer #2 (Remarks to the Author):

This study's strengths remain solid as described in the previous review and merit publication. The authors took considerable effort to modify calculations and description to make them more understandable. Two significant problems remain which must be addressed prior to publication. The first is an error reporting the analysis results. The second is a misunderstanding and representation of drainage failures.

1. Design return periods versus the likelihood of failure. The use of design return period was not addressed. The use of changes to design return periods as an equivalent metrics as changes in the likelihood of failure (exceedance probability) in any given year is incorrect.

For example in the abstract. "global transportation assets are expected at least a 25% increase in their likelihood of experiencing an extreme rainfall event that exceeds their design standards, which may increase to 69.9% under ~four degrees of warming by late-21st century."

And on page 12

"When considering a safety factor for climate change adaptation, the transportation assets are expected to maintain their designed risk level in the future."

And again in the conclusion there is the use of both return periods and increased frequencies in the same sentence. "Our results show that about 97% of global transportation assets will be exposed to precipitation with higher frequency return periods, with increased frequencies between 0 and 50% for both RCP4.5 and RCP8.5 scenarios."

The math does not work out this way. See the attached supplement for details on the relationship between the return period and the exceedance prob that correspond as well as changes to both.

The authors need to address the error in an appropriate manner. Unfortunately, their response to the previous review is "If there is a 50% decreases in return period, it means the likelihood that such event occurs has doubled." is incorrect and conveys their lack of understanding of the relationship between return periods and exceedance probabilities. Without this correction, the work should not be published because it is wrong.

The authors could correct all the cases where exceedance probability or risk of failure is stated or implied to return period changes. This would considerably weaken the overall findings and very much seems to go against the main point of the analyses conducted in this paper as well as the main finding: that globally drainage systems are under designed for future conditions.

2. Drainage and impacts to infrastructure

The authors have missed the point and a broader understanding of the importance of drainage performance. First, most of the time when the rainfall event exceeds the design threshold there is no impact at all. It might back up a bit over the culvert but there is no erosion, no failure. Water resources engineers working in transportation will readily tell you that this is the case. Just considering the return periods and a simple thought experiment... in New York, if highways are designed for a 10-yr event and local roads for a 5-yr event, then in any given year 10% of the highways should fail and 20% of the local roads should fail. That absolutely does not happen. Failures are very very infrequent. This is likely because of two reasons. The first is that most drainage systems already have a fairly significant safety factor. The second is that "failure" does not mean that the system breaks, but only that the pipe does not pass all the water without backing up a bit or the drainage system doesn't provide the treatment as required.

Another point that is missed is that the examples of failures on page 2 of the introduction are NOT drainage failures but failures due to large floods (e.g., 25, 50, 100 year or beyond) in which the design failure was not a drainage design failure but a riverine flood design issue. For this point, there is an implication throughout the manuscript that transportation systems will catastrophically fail in the future. This work examines changes in low magnitude/high frequency floods.

From the author's response to point 6 Infrastructure and in particular drainage systems are almost always oversized due to "rounding up" to larger nominal pipe sizes. Failure of these systems does not mean damage except in very few cases – failure would be a bit of water backing up. Realistic representation of the implications of "failure" is missing and overstated.

Response: We thank for reviewer's comment. In this study, we focus on the transportation infrastructure exposure due to inadequate designed drainage. When the precipitation exceeds the design threshold, the excess water (water backing up could be one of the reasons) will adversely affect the transport assets, such as cut or fill slope failures, road surface erosion, etc. While the damage of drainage pipe is not considered in this study. To avoid the confusion, we have added the following description in lines 107-108, page 5:

Global transportation infrastructure exposure to the change of precipitation in a warmer world

Manuscript ID: NCOMMS-22-02358A

We would like to thank the editor and reviewers for their thorough reading of the revised manuscript and the valuable remarks that helped us to improve the manuscript. We have revised the manuscript carefully according to the reviewers' comments, and have incorporated the suggestions into the revised manuscript.

The notes below provide a point-by-point response to each comment from the referees. The texts with blue font are the reviewer's original comments, and the texts with black font are the authors' responses. If there is any question addressed unclearly or unsatisfied, we are always willing to make a third revision based on the reviewer's comments. Thank you again for the opportunity to be considered for publication in *Nature Communications*.

Reviewer #1

The authors have done a fine job in addressing reviewer comments and improving this paper, and I do commend them for this effort! I have a couple follow up points that they might wish to address.

1. I am still struggling with the absolute exposure metric and what it tells us (and how you discuss the results). I completely understand its use. However, it simply appears to be dominated by infrastructure density (and country size) at the top (no surprises here), while as one moves down the list I guess extreme weather and other factors starts playing a part? I can only speculate as these dynamics and their nuances, and how relative exposure compares and relates, are not discussed. It would really help to discuss these results further and more deeply, to provide illumination.

Response: We thank for the reviewer's helpful comments. As described in Section 2.2, the absolute exposure is affected by two factors: the spatial distribution of the assets and the change in the design return period at a given location. From the results, it seems that in general the infrastructure density dominates, however, when we look at the ranking of each country, we can see that both the density and precipitation change contributes. For example, Sweden ranks 20th for total road and railway assets, but ranks 9th for absolute exposure in the mid-century under RCP4.5 scenario; While Greece ranks 39th for total road and railway assets, but 84th for absolute exposure in the mid-century under RCP4.5 scenario. To make it clear, in the revised

version we provide a separate supplementary table for rankings of all countries in total assets, absolute exposure and relative exposure under different scenarios (Supplementary Data). We also added the following descriptions in lines 134-139, page 7:

Although the infrastructure density dominates the absolute exposure, when we further investigate the rankings of countries (see supplementary data), we find that both the infrastructure density and the precipitation change contribute. Sweden and Greece, for example, are ranked respectively 20th and 39th place for the total length of road and railway assets. Under the mid-century RCP4.5 scenario, however, their rankings are 9th (Sweden) and 84th (Greece) in absolute exposure, unveiling the contribution of precipitation change.

Compared to the results of absolute exposure, precipitation change plays a more important role in relative exposure. The benefit of using relative exposure is that it can highlight countries with fewer assets but are strongly affected by precipitation change. These countries are usually less developed countries or small countries and may be overseen in absolute exposure due to low transportation density. To make it clear, we have added the following description in lines 149-152 and lines 158-163, page 8:

Compared to absolute exposure, the relative exposure can highlight countries with fewer assets but are strongly affected by precipitation change. These countries are usually less developed and may be overseen when exploring the results for absolute exposure due to their relatively low amount of transportation assets.

Interestingly, we do find some countries experience a low absolute exposure but a high relative exposure. In Panama, for example, 99% of the grids will face an exposed ratio of over 80% in the late-21st century under the RCP8.5 scenario, making a large proportion of the transportation infrastructure in Panama highly vulnerable to rising temperatures. Yet, Panama only has a total length of assets of 6539 km and ranks 107th in absolute exposure in the late-21st century under the RCP8.5 scenario (see supplementary data).

2. The authors added: "Furthermore, understanding the impact of climate change on infrastructure assets is crucial to design adaptation strategies for local governments and municipalities. Our study provides a picture of future risks faced by transportation infrastructure and helps bridge the knowledge gap between climate science communities, infrastructure communities, and policy makers. This knowledge can also incentivize local governments or municipalities to design infrastructure adaptation measures to cope with the risk of anthropogenic warming." These statements are rather obvious and are not new or insightful observations. As far as I can see, this work does not offer tangible and direct guidance to local governments, who need more detailed and context-sensitive advice. I would suggest

either removing any mention of impact to local authorities, or considering further exactly how this work directly impacts them.

Response: We thank for the reviewer's comment. We agree that this may be too far-fetched. We have now removed these final sentences. Our study indeed helps with identifying hotspots globally, which can be used to target "deep-dive studies" to better assess the local or regional situation. This study indeed provides a starting point, but is not an end point for local authorities.

Reviewer #2

This study's strengths remain solid as described in the previous review and merit publication. The authors took considerable effort to modify calculations and description to make them more understandable. Two significant problems remain which must be addressed prior to publication. The first is an error reporting the analysis results. The second is a misunderstanding and representation of drainage failures.

1. Design return periods versus the likelihood of failure. The use of design return period was not addressed. The use of changes to design return periods as an equivalent metrics as changes in the likelihood of failure (exceedance probability) in any given year is incorrect.

For example in the abstract. "global transportation assets are expected at least a 25% increase in their likelihood of experiencing an extreme rainfall event that exceeds their design standards, which may increase to 69.9% under ~four degrees of warming by late-21st century."

And on page 12

"When considering a safety factor for climate change adaptation, the transportation assets are expected to maintain their designed risk level in the future."

And again in the conclusion there is the use of both return periods and increased frequencies in the same sentence. "Our results show that about 97% of global transportation assets will be exposed to precipitation with higher frequency return periods, with increased frequencies between 0 and 50% for both RCP4.5 and RCP8.5 scenarios."

The math does not work out this way. See the attached supplement for details on the relationship between the return period and the exceedance prob that correspond as well as changes to both.

The authors need to address the error in an appropriate manner. Unfortunately, their response to the previous review is "If there is a 50% decreases in return period, it means the likelihood that such event occurs has doubled." is incorrect and conveys their lack of understanding of the relationship between return periods and exceedance probabilities. Without this correction, the work should not be published because it is wrong.

The authors could correct all the cases where exceedance probability or risk of failure is stated or implied to return period changes. This would considerably weaken the overall findings and

very much seems to go against the main point of the analyses conducted in this paper as well as the main finding: that globally drainage systems are under designed for future conditions.

Response: We thank for the reviewer’s comment and apologize for the inappropriate reporting. We totally agree with the reviewer that changes to design return periods are not as an equivalent metrics as changes in the likelihood of failure (exceedance probability), but it is true that we used the wrong expression. In our response to the previous review, we tried to express that if there is a 50% decrease in return period, it means the exceedance probability of such an event has doubled. This is true for a 50% decrease in return period which corresponds to a 100% increase in exceedance probability, i.e., $1 / (RP \times (1 - 50\%))$ leads to a doubled exceedance probability. But we do make a mistake in the abstract in describing the likelihood change of 25% decrease in return period. Furthermore, the term of likelihood could lead to a misunderstanding to readers. In the revised manuscript, we have changed ‘likelihood’ to ‘exceedance probability’. To make it clear, we also provide a supplementary table to illustrate the relationship between return period change and exceedance probability change. Our specific revisions are given below. Thank you again for pointing out this error! While these corrections indeed will not impact our main findings, we believe the important conclusions of our manuscript still hold.

Table S4. Relations between return period change and exceedance probability change

Decrease of Return Period	Increase of Exceedance Prob
10%	11%
15%	18%
20%	25%
25%	33%
30%	43%
35%	53%
40%	67%
45%	82%
50%	100%

- 1) Page1, Line8-11: *Under ~two degrees of warming in mid-century (RCP 8.5 scenario), 43.6% of the global transportation assets are expected to experience at least a 25% decrease in design return period of extreme rainfall (a 33% increase in exceedance probability), which may increase to 69.9% under ~four degrees of warming by late-21st century.*
- 2) Page2-3, Line48-53: *Nearly 88.4%–92.2% of global road and railway assets will face a decrease in return periods under the Representative Concentration Pathway 4.5 (RCP4.5 scenario) scenario; of these, 58.7% and 73.8% (~1.8 degree of warming in mid-21st century/~2.5 degree of warming in late-21st century, respectively) of assets will*

experience a decrease of return period larger than 25% (e.g., what used to be a 1-in-10-year event for the period of 1971–2000 becomes a less than 1-in-7.5-year event in the corresponding simulation), corresponding to an increase in exceedance probability of more than 33%.

- 3) Page13-14, Line249-253: *Our results show that approximately 88.4% (mid-21st century, RCP4.5 scenario)–94.6% (late-21st century, RCP8.5 scenario) of global transportation assets will be exposed to precipitation with increased frequencies. Under the RCP4.5 scenario, a total of 6.8 million km (mid-21st century) and 11.0 million km (late-21st century) of global transportation assets will be exposed to more frequent extreme precipitation (decrease in design return period larger than 25%).*

2. The authors have missed the point and a broader understanding of the importance of drainage performance. First, most of the time when the rainfall event exceeds the design threshold there is no impact at all. It might back up a bit over the culvert but there is no erosion, no failure. Water resources engineers working in transportation will readily tell you that this is the case. Just considering the return periods and a simple thought experiment... in New York, if highways are designed for a 10-yr event and local roads for a 5-yr event, then in any given year 10% of the highways should fail and 20% of the local roads should fail. That absolutely does not happen. Failures are very very infrequent. This is likely because of two reasons. The first is that most drainage systems already have a fairly significant safety factor. The second is that “failure” does not mean that the system breaks, but only that the pipe does not pass all the water without backing up a bit or the drainage system doesn’t provide the treatment as required. Another point that is missed is that the examples of failures on page 2 of the introduction are NOT drainage failures but failures due to large floods (e.g., 25, 50, 100 year or beyond) in which the design failure was not a drainage design failure but a riverine flood design issue. For this point, there is an implication throughout the manuscript that transportation systems will catastrophically fail in the future. This work examines changes in low magnitude/high frequency floods

From the author’s response to point 6 Infrastructure and in particular drainage systems are almost always oversized due to “rounding up” to larger nominal pipe sizes. Failure of these systems does not mean damage except in very few cases – failure would be a bit of water backing up. Realistic representation of the implications of “failure” is missing and overstated.

Response: We thank for the reviewer’s helpful comments. We agree with the reviewer that when a rainfall event exceeds the design threshold of the drainage system, it does not mean the transport assets will fail. But it could lead to road usage disruptions and could cause a further

potential deterioration of transport assets by years. As discussed in Ref [1-3], excessive water can cause traffic safety, erosion, reduced bearing capacity in the subgrade, and reduced pavement life time and increased pavement management costs. As for railways, it is pointed that 92% of stability problems in French railway network have been related to insufficient drainage of the platforms ^[4].

[1]<https://www.roadex.org/e-learning/lessons/drainage-of-low-volume-roads/introduction-why-drainage-is-important/>

[2]<https://www.engineeringenotes.com/highway-construction/highway-drainage-need-and-types-of-highway-drainage-system/48795>

[3] <https://www.aboutcivil.org/importance-of-drainage.html>

[4] Roberto Sañudo et al. Drainage in railways. Construction and Building Materials. 2019, 210:391-412.

In the revised manuscript, we have avoided the use of ‘failure’, and have clarified the meaning of ‘damage’ and ‘absolute exposure’ as suggested by the reviewer in lines 96-98, page 5:

When the probability of a rainfall event exceeds this design threshold, the excess water may adversely affect the transport assets. This could cause disruption of its usage, or structural degradations such as road surface erosion, reduced bearing capacity, and shortened structural life.

Also in the Methods part in lines 360-361, page 18:

We assumed that, when there was a more than 25% decrease in the design return period, the transportation infrastructure could be affected by the inadequate drainage condition.

As for examples given in the introduction part, we agree with the reviewer’s suggestion that they mostly focused on low probability events, whereas the manuscript focuses on high-probability events. As such, we have removed them in the revised manuscript.